# What should a neuron aim for? Designing local objective functions based on information theory

**Andreas C. Schneider**[1,2,*], **Valentin Neuhaus**[1,2,*], **David A. Ehrlich**[3,2,*], **Abdullah Makkeh**[3,2], **Alexander S. Ecker**[4,2], **Viola Priesemann**[2,1,†], **Michael Wibral**[3,2,†]

[1]Faculty of Physics, Institute for the Dynamics of Complex Systems, University of Göttingen
[2]Max Planck Institute for Dynamics and Self-Organization, Göttingen
[3]Göttingen Campus Institute for Dynamics of Biological Networks, University of Göttingen
[4]Institute of Computer Science and Campus Institute Data Science, University of Göttingen
`{andreas.schneider, valentin.neuhaus, viola.priesemann}@ds.mpg.de`
`{davidalexander.ehrlich, michael.wibral}@uni-goettingen.de`

## Abstract

In modern deep neural networks, the learning dynamics of individual neurons are often obscure, as the networks are trained via global optimization. Conversely, biological systems build on self-organized, local learning, achieving robustness and efficiency with limited global information. Here, we show how self-organization between individual artificial neurons can be achieved by designing abstract bio-inspired local learning goals. These goals are parameterized using a recent extension of information theory, Partial Information Decomposition (PID), which decomposes the information that a set of information sources holds about an outcome into unique, redundant and synergistic contributions. Our framework enables neurons to locally shape the integration of information from various input classes, i.e., feedforward, feedback, and lateral, by selecting which of the three inputs should contribute uniquely, redundantly or synergistically to the output. This selection is expressed as a weighted sum of PID terms, which, for a given problem, can be directly derived from intuitive reasoning or via numerical optimization, offering a window into understanding task-relevant local information processing. Achieving neuron-level interpretability while enabling strong performance using local learning, our work advances a principled information-theoretic foundation for local learning strategies.

## 1 Introduction

Most artificial neural networks (ANNs) optimize a single global objective function through algorithms like backpropagation, orchestrating parameters across all computational elements of the network as a unified computational structure. While this global objective approach has proven effective across a large variety of tasks, the role of the individual neuron often remains elusive, hindering the understanding of how local computation contributes to global task performance.

In contrast, biological neural networks exhibit a markedly different approach to learning, relying strongly on self-organization between neurons. Further, biological neurons must individually learn according to information that is locally available, because of being limited by physical constraints regarding locality and communication bandwidth. Interestingly, these limitations come with the advantage that computation of a neuron can also be interpreted locally, at least in principle. Combining this local interpretability with the observation that biological neural networks achieve high levels of efficiency and robustness even in the absence of a centrally coordinated objective raises the question: Could ANNs benefit from similar local learning mechanisms to enhance the *local* interpretability of their computations, while maintaining performance?

---

*Equally contributing first authors;†Equally contributing last authors

Neuroscience models have indeed shown that local learning rules (Hebb, 1949; Dan and Poo, 2004; Song et al., 2000) can solve a variety of tasks, such as sequence learning, unsupervised clustering, and classification tasks (Hebb, 1949; Földiak, 1990; Cramer et al., 2020). Often formulated on the basis of temporal coincidence (spike-timing), these very mechanistic rules do not offer direct insights into the actual information processing at each neuron. To facilitate such functional insight, more recently *information theory* has been used to formulate more abstract, but interpretable frameworks for local learning rules (e.g., Kay 2000; Wibral et al. 2017).

Information theory (Shannon, 1948) allows one to abstractly prescribe how much of the information from different inputs should be conveyed to the output of a neuron (Kay, 2000). Moreover, it provides a general and flexible mathematical framework, which captures the information processing at an abstract and interpretable level free of details of the exact implementation. As such, information theory has been employed to define general optimization goals, including global objectives like cross-entropy loss (Goodfellow et al., 2016) and more localized ones such as local greedy infomax (Löwe et al., 2019), and early attempts at passing only information that is coherent between multiple inputs (Kay and Phillips, 2011).

Nevertheless, classical information theory falls short of describing how multiple sources work together to produce an output. In fact, it completely lacks the capacity to describe how the information from different sources is integrated in redundant, synergistic or unique ways in the output. These different ways, or information *atoms*, however, contribute very differently to a neuron's function: Redundant information quantifies only coherent information from multiple sources, for example to have a neuron forward only information that is in agreement between sensory input and internal context information; while unique information reflects what can be obtained from a single source but not from others, for example to make neurons not encode the same information that their peers already encode. Lastly, synergy describes the synthesis of new information from taking multiple sources into account simultaneously, for instance when comparing two signals and computing a difference or error signal.

This complexity of operations on input information from multiple classes of sources can be quantified by using the recent framework of "Partial Information Decomposition (PID)" (Williams and Beer, 2010; Gutknecht et al., 2021; Makkeh et al., 2021). Given multiple inputs to an output, it can dissect the mutual information into different information atoms, which enumerate the different ways in which the sources can interact to produce the target, as explained above. Through this decomposition, one can paint a comprehensive, yet interpretable picture of how the different input variables contribute to the local computation of an individual neuron.

PID has already been utilized to analyze the information representation and flows in DNNs (Ehrlich et al., 2023; Tax et al., 2017). We show how this approach can be inverted by using PID to directly formulate goal functions on the individual neuron level. These "infomorphic neurons" then have the ability to optimize for encoding specific parts of the information they receive from their inputs, allowing for an application to tasks from supervised, unsupervised and memory learning, as demonstrated explicitly in Makkeh et al. (2025). However, neurons with only two different inputs struggle to achieve good performance on classification tasks. It seems that in general, three distinct signals are necessary for local neuron learning: A receptive signal providing the input, a relevance signal helping to filter this input and a distribution signal enabling self-organization between neurons. In classical neural networks, these different classes of information are provided implicitly via the global gradient signals of backpropagation.

In this work, we show how ANNs based on infomorphic neurons with three input classes can be used to solve supervised classification tasks with a performance comparable to backpropagation, despite relying only on locally available information. We demonstrate how the parameters for the objective functions can either be determined from intuitive reasoning or be optimized by hyperparameter optimization techniques. The latter approach then sheds light onto the *local* information processing needed to solve a complex task. Through this approach, we demonstrate that local learning rules rooted in information theory can provide an understanding of local ANN learning while maintaining their performance.

The main contributions of this work are as follows: Firstly, we use information theory, and in particular PID, to formulate interpretable, per-neuron goal functions with three input classes, significantly advancing the capabilities of the approach originally presented in Makkeh et al. (2025). Secondly, we systematically optimize the goal function parameters for these "infomorphic" neurons, provide

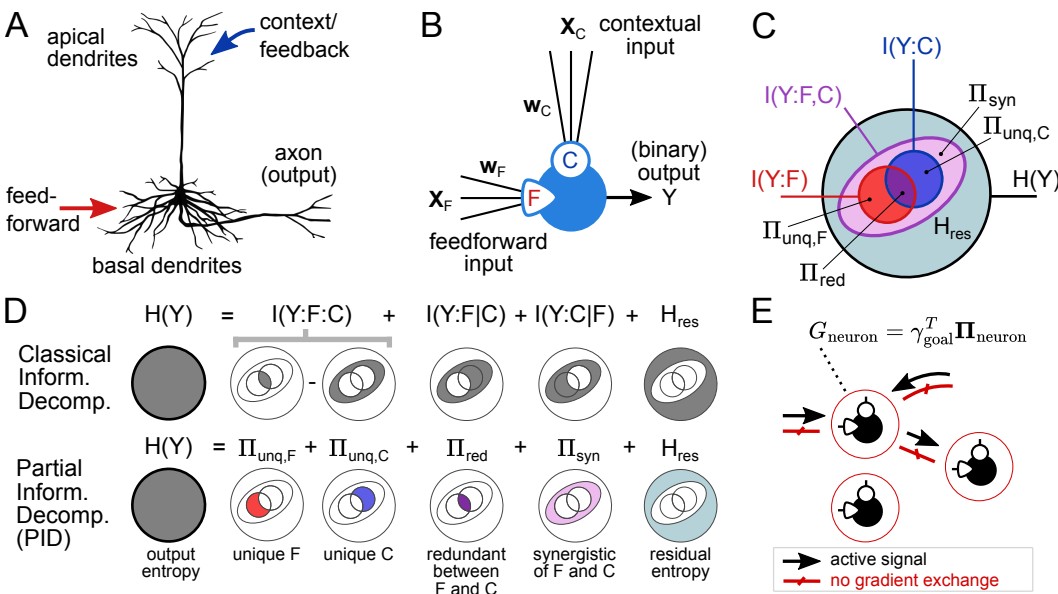

Figure 1: **Infomorphic neurons are abstract, information-theoretic neurons inspired by the structure of pyramidal neurons (Wibral et al., 2017). They are trained by adjusting their synaptic weights according to a PID-based goal function. A,B**. Inspired by the distinction between apical and basal dendrites in cortical pyramidal neurons, infomorphic neurons are defined as computational units with separate feedforward ($F$) and contextual ($C$) input classes. **C**. Partial information decomposition (PID) allows one to dissect the total entropy of the neuron into explainable components. **D**. PID enables one to distinguish how much information comes *uniquely* from either the feedforward $F$ ($\Pi_{\text{unq,F}}$) or contextual $C$ input ($\Pi_{\text{unq,C}}$) and how much they contribute *redundantly* ($\Pi_{\text{red}}$), or *synergistically* ($\Pi_{\text{syn}}$). Classic information theory (top) cannot disentangle these information atoms: The classic entities cover several of the atoms, so that effectively one can only measure redundant minus synergistic information. **E**. Formulating goal functions $G_{\text{neuron}}$ in terms of PID-atoms $\Pi_{\text{neuron}}$ enables one to formulate how strongly redundant, unique, or synergistic information should contribute to a neuron's output. Figure adapted from Makkeh et al. (2025).

an intuitive interpretation, and thereby obtain insights into the local computational goals of typical classification tasks. Thirdly, for classification tasks we show that PID-based learning can achieve performance comparable to backpropagation despite being limited to local information only, while being interpretable on a per-neuron basis.

Next, we give a succinct account of infomorphic neurons with two input classes following Makkeh et al. (2025) before generalizing the framework to the more capable version with three input classes in Section 3.

## 2 BIVARIATE INFOMORPHIC NEURONS

Neurons can be viewed as information processors that receive a number of input signals and process them to produce their own output signal. From an information-theoretic perspective, the output $Y$ of a neuron can be considered a random variable, and the total output information can be quantified using the Shannon entropy $H(Y)$. From an information channel perspective, this output information consists of two parts: The mutual information $I(Y : \boldsymbol{X})$ coming from the inputs $\boldsymbol{X}$ and the residual entropy $H(Y|\boldsymbol{X})$ arising from stochastic processes within the neuron.

Mutual information can be used to quantify the amount of information that is carried by different input classes about the neuron's output by considering the aggregated feedforward ($F$), contextual ($C$) and lateral inputs ($L$) as sources $X$ individually. Nevertheless, classical information theory cannot quantify how much of the information is provided redundantly, uniquely or synergistically by different inputs, making it impossible to describe *how exactly* the information from two source

variables contribute to creating the output. Using PID, the total mutual information $I(Y : F, C)$ between the output of the neuron $Y$ and two aggregated inputs, e.g., $F$ and $C$, can be dissected into four PID *atoms* (see Fig. 1.**C**): The *unique* information of the feedforward connection about the output $Y$, $\Pi_{\mathrm{unq,F}}$, is the information which can only be obtained from the feedforward and not the context input, with the unique information $\Pi_{\mathrm{unq,C}}$ of the context being defined analogously. The *redundant* information $\Pi_{\mathrm{red}}$ reflects the information which can equivalently obtained from either feedforward or contextual inputs about $Y$, while finally the *synergistic* information $\Pi_{\mathrm{syn}}$ can only be obtained from both inputs considered jointly. All classical mutual information terms between the target and subsets of source variables can be constructed from these PID atoms through the consistency equations (Williams and Beer, 2010)

$$
\begin{aligned}
I(Y : F, C) &= \Pi_{\mathrm{red}} + \Pi_{\mathrm{unq,F}} + \Pi_{\mathrm{unq,C}} + \Pi_{\mathrm{syn}}, \\
I(Y : F) &= \Pi_{\mathrm{red}} + \Pi_{\mathrm{unq,F}} \text{ and} \\
I(Y : C) &= \Pi_{\mathrm{red}} + \Pi_{\mathrm{unq,C}}.
\end{aligned}
\tag{1}
$$

However, these atoms are underdetermined as there are four unknown atoms with only three consistency equations providing constraints. For this reason, an additional quantity needs to be defined, which is usually a measure for redundancy (Williams and Beer, 2010). Such a definition for redundancy, based on the concept of shared exclusions in probability space, is given by the shared-exclusion PID, $I_{\cap}^{\mathrm{sx}}$ (Makkeh et al., 2021) (see Appendix A.2). Importantly, the resulting PID is differentiable with respect to the underlying probability mass function, which allows for a specification and subsequent optimization of learning goals based on PID.

Depending on the task the network is set to solve, different PID atoms become relevant to the information processing. In this work, we focus on the application to supervised learning tasks, in which the ground-truth label is provided as the context $C$ during training. Here, the intuitive goal for the neuron is to foster redundancy between the feedforward and context inputs in its output, to capture only the feedforward signal that agrees with the label. Likewise, if the goal is unsupervised encoding of the input, lateral connections between neurons can be used as the context signal and the goal might become maximizing the unique information of the feedforward input about the output, i.e., to capture only the information which is not already encoded in other neurons (Makkeh et al., 2025).

PID can not only be used to describe the local information processing, but also to optimize it through the maximization of a PID based objective function. Generally, such an objective function can be expressed as a linear combination of PID atoms as

$$
G = \gamma_{\mathrm{red}}\Pi_{\mathrm{red}} + \gamma_{\mathrm{unq,F}}\Pi_{\mathrm{unq,F}} + \gamma_{\mathrm{unq,C}}\Pi_{\mathrm{unq,C}} + \gamma_{\mathrm{syn}}\Pi_{\mathrm{syn}} + \gamma_{\mathrm{res}}H_{\mathrm{res}} = \boldsymbol{\gamma}^T\boldsymbol{\Pi}, \tag{2}
$$

where the residual entropy $H_{\mathrm{res}} = H(Y|F,C)$, is included in the vector $\boldsymbol{\Pi}$ for brevity of notation. As the $I_{\cap}^{\mathrm{sx}}$ measure is differentiable, the goal function can be optimized by adjusting the weights of the incoming connections of the inputs $F$ and $C$ using gradient ascent—with the analytic formulation of the gradients given in Makkeh et al. (2025), or using a suitable autograd algorithm.

While such infomorphic neurons can be used for supervised learning tasks when each neuron is provided a part of the label to focus on (Makkeh et al., 2025), two input classes are insufficient for fully self-organizing the necessary local information processing. This is because if each neuron is tasked with encoding information that is redundant between the whole label and the feedforward signal with no regard to the other neurons, they will inevitably start encoding much of the same information, leading to poor performance. To avoid this, the neurons need to be made aware of what the other neurons encode, similar to the unsupervised example. This can be achieved by incorporating a third input class for lateral connections between neurons of the same layer, as presented next.

## 3 TRIVARIATE INFOMORPHIC NEURONS

Including a third source variable results in a so-called trivariate PID which is more complex than the bivariate case, with atoms in general now describing redundancies between synergies. For example, the atom $\Pi_{\{F\}\{C\}}$ denotes information that can be obtained redundantly from the feedforward and context signal but is unavailable from the lateral inputs, making it 'uniquely redundant' to $F$ and $C$. Likewise, the atom $\Pi_{\{F\}\{C,L\}}$ denotes information that can be obtained from $F$ alone but also synergistically from the pair $C, L$ (but not from $C$ or $L$ alone). Overall, upon the introduction of a third source, the four PID atoms of the bivariate setup turn into a total of 18 trivariate PID

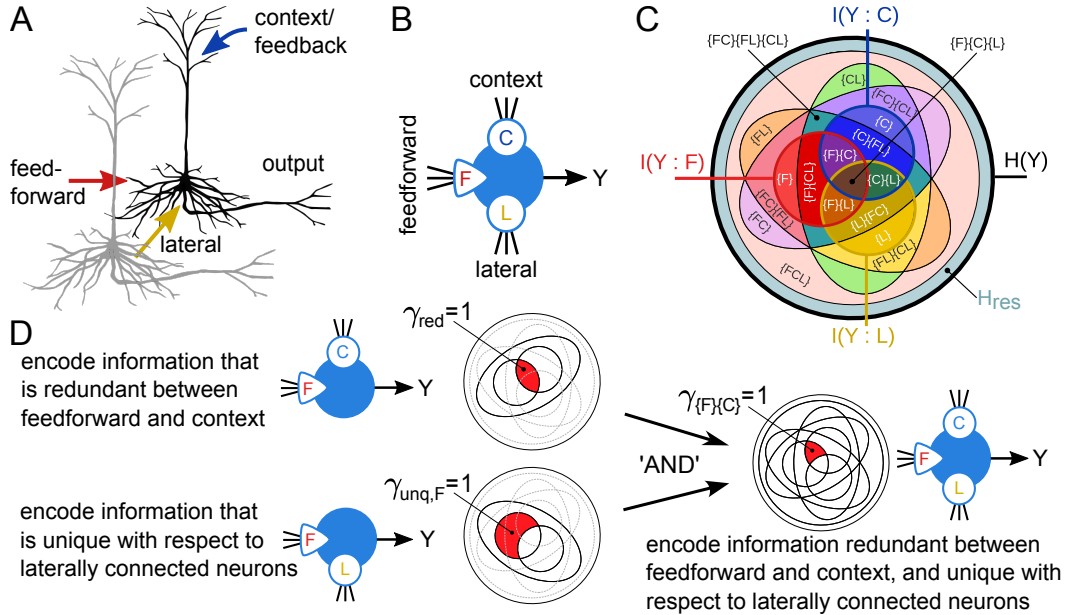

Figure 2: **Adding lateral connections as a third input class enables neurons to self-organize to encode relevant and unique information. A,B.** Infomorphic neuron with three inputs - namely feed-forward ($F$), contextual ($C$) and lateral ($L$) inputs. **C.** With three input classes, the number of PID-atoms increases to 18, represented by different colors, plus the residual entropy $H_{\text{res}}$ in the outer circle. Classical information-theoretic quantities such as the entropy $H(Y)$ and mutual information $I$ with individual sources are depicted by ovals, indicating how they can be built from PID atoms. **D.** Three input classes allow for the optimization of more complex goals based on 19 different terms (compared to only five in Fig. 1.**C**). This trivariate PID allows to combine two bivariate objective functions: In a supervised learning task, the intuitive goal is to maximize information in the neuron's output that is redundant between the feedforward input $F$ and label $C$, while simultaneously ensuring the neuron's output stays unique with respect to lateral neurons $L$. While bivariate goal functions only allow for optimizing one of these objectives at a time, both objectives can be combined to the goal of maximizing the single atom $\Pi_{\{F\}\{C\}}$ in the trivariate case.

atoms (Gutknecht et al., 2021) as shown in Fig. 2.**C**. These PID atoms $\Pi_\alpha$ can in general be addressed by their *antichains* $\alpha$, i.e., sets of sets of input variables, with the inner sets describing synergies and the outer set redundancies between them (Gutknecht et al., 2021).

Including the residual entropy $H_{\text{res}} = H(Y|F,C,L)$, the decomposition vector $\mathbf{\Pi}$ and goal parameter vector $\boldsymbol{\gamma}$ thus have 19 elements each. The general goal function of a neuron is again given by $G = \boldsymbol{\gamma}^T \mathbf{\Pi}$ and can be set and optimized in the same way as in the bivariate case:

$$G = \gamma_{\{F\}\{C\}\{L\}}\Pi_{\{F\}\{C\}\{L\}} + ... + \gamma_{\{FCL\}}\Pi_{\{F,C,L\}} + \gamma_{\text{res}}H_{\text{res}} = \boldsymbol{\gamma}^T \mathbf{\Pi}. \tag{3}$$

Utilizing the expressiveness and interpretability of trivariate PID, heuristic goal functions can be derived for different classes of tasks from intuitive reasoning. For the supervised learning task to be solved here, the introduction of the lateral connections as a third input allows to synthesize an intuitive goal function from the two bivariate goals introduced before (see Fig. 2.**D**). As outlined in the bivariate case above, canonical goals might either be to foster redundancy between the feedforward and context signals with no regards for what lateral neurons do, or, alternatively, to optimize for uniqueness of the output with respect to the lateral inputs, which does not differentiate between task relevance of information. With trivariate infomorphic neurons, however, these goals can simultaneously be achieved by optimizing for the atom $\Pi_{\{F\}\{C\}}$, which represents the redundancy of the feedforward and context signal which is at the same time unique with respect to the lateral inputs. In simple terms, this enables the neuron to find coherent information between label and input data, that is not yet encoded by other neurons.

## 4 EXPERIMENTS

We now explain how the general concept of trivariate infomorphic neurons can be practically utilized to construct networks solving a supervised classification task, i.e., the MNIST supervised handwritten digit classification task. To this end, we demonstrate how an infomorphic network with one hidden layer can be set up using either heuristic or optimized goal function parameters.

**Neuron structure and activation function** To construct infomorphic networks, a number of practical decisions need to be made for how the output signal is constructed from the inputs in the forward pass. As a first step, the higher-dimensional signals from the different input classes are aggregated to the single numbers $F$, $C$ and $L$ by a weighted sum. Subsequently, these aggregated inputs are passed into an activation function, which can be chosen arbitrarily, but needs to fit the requirements of the specific application. For the supervised classification task at hand, the function $A(F, C, L) = F[(1 - \alpha_1 - \alpha_2) + \alpha_1 \sigma(\beta_1 FC) + \alpha_2 \sigma(\beta_2 FL)]$ has been chosen, where $\sigma$ refers to the sigmoid function and $\alpha_1 = \alpha_2 = 0.1$ and $\beta_1 = \beta_2 = 2$ are parameters that shape the influence of the input compartments on the activation function. This activation function builds on concepts of Kay (1994) and makes a clear distinction between the driving feedforward input and the modulatory context and lateral inputs, ensuring that the network performs similarly during training, when the context inputs is provided, and for evaluation, where it is withheld (i.e., by setting $C = 0$). To show the independence of the infomorphic approach from a particular activation function, the results are additionally compared to a simple linear activation function $A(F, C, L) = F + 0.1L + 0.1C$. The output of the activation function is finally mapped to the unit interval by a sigmoid function, whose output is used as a probability to stochastically output "1" (firing) or "-1" (non-firing). For the bivariate output layer, the sigmoid is not sampled from but just interpreted as a firing probability, with the neuron with the highest value being used as the network prediction. The lateral connections between neurons add recurrence to the network, requiring for the same input to be shown multiple times to ensure convergence of the activations. In practice, however, we have seen that presenting the same input twice is sufficient to achieve proper network function (see Appendix A.9). The inference procedure for infomorphic networks is summarized in pseudocode in Appendix A.1.

**Discretization and local gradients** To optimize a PID-based goal function during training, the first step is to estimate the PID atoms. Since the original $I_\cap^{\text{sx}}$ measure only works with discrete variables, the aggregated inputs from a batch of samples are first discretized to 20 levels each (see Appendix A.11). Subsequently, the empirical probability mass function is constructed from the batch, from which the PID atoms are computed by means of a plug-in estimation. Finally, the current value of the goal function can determined as a linear combination of PID atoms (Algorithm 2 and Algorithm 3 in Appendix A.1). For the local gradient computation, the gradient of the individual neuron's objective function is backpropagated via the output to produce the neuron's weight updates.

**Choosing goal parameters** The parameters $\gamma$ for a PID goal function can be obtained in two distinct ways—either heuristically from intuitive notions about the nature of the computations necessary to solve the problem, as explained before, or optimized using a suitable hyperparameter optimization procedure. In this work, we contrast both approaches. For the hyperparameter optimization, we compared two established techniques: The Tree-structured Parzen Estimator (TPE, Bergstra et al., 2011), that splits the sample points into two groups with high and low performance, fits a model to each group and draws the next sample points according to the ratio of probabilities, and the Covariance Matrix Adaptation Evolution Strategy (CMA-ES, Hansen, 2016), which generates new evaluation points by sampling from a multivariate Gaussian distribution that was fitted to the best samples from previous iterations.

**Network Setup** To solve the MNIST task (LeCun et al., 1998), we devised a network architecture consisting of an input layer, a hidden layer made up from $N_{\text{hid}}$ trivariate infomorphic neurons and an output layer consisting of 10 bivariate infomorphic neurons. The trivariate neurons in the hidden layer each receive the whole image as their feedforward signal, a context signal that guides their learning, and the output of other neurons of the same layer through lateral connections. The bivariate neurons in the output layer receive all outputs of the hidden layer and additionally exactly one element of the one-hot label as context, which is only provided during training, and are trained to maximize redundancy between their input and the information about the one label they observe. We explored three slightly different approaches to how the hidden layer can be set up using trivariate infomorphic neurons, illustrated in Fig. 3.**A**. In setup 1, we use the fully connected one-hot label as

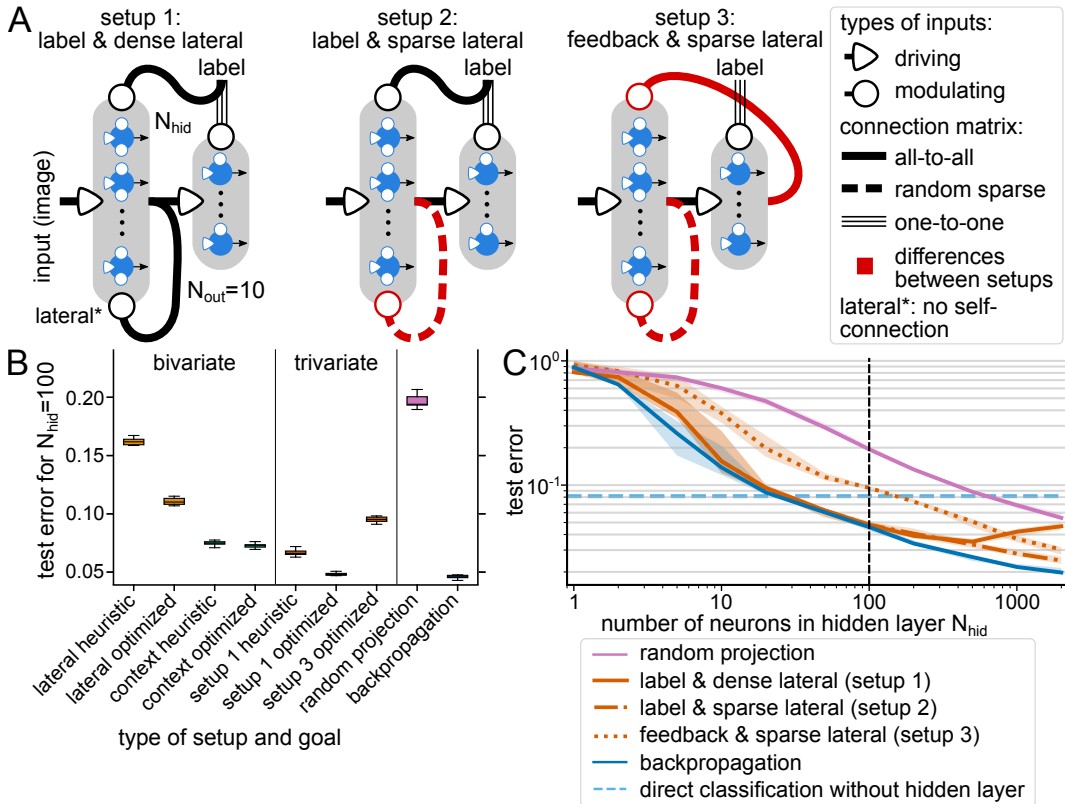

Figure 3: **Infomorphic networks with three input classes approach the performance of a similar network trained with backpropagation and outperform networks with two input classes on the MNIST handwritten digit classification task.** **A**. In the hidden layer of the network, the infomorphic neurons receive feedforward and lateral connections as well as either the ground-truth label or feedback from the output layer, outlined in three setups. **B**. Networks with trivariate infomorphic neurons outperform models with bivariate neurons with both, heuristic and optimized goals for a hidden layer size of 100 neurons. Only the setup with a feedback signal instead of the label performs worse for this layer size but outperforms the bivariate models for larger hidden layers as shown in Fig. 8. **C**. Infomorphic networks achieve similar performance as the same network trained with backpropagation. For larger layers, using a sparse connectivity significantly improves performance (setup 2). The lines indicate mean values, with the intervals depicting the maximum and minimum of 10 runs. The goal function parameters have been optimized for networks with a hidden layers size of 100 neurons, indicated by the dashed line in **C**.

$C$ and the output of all neurons of the same layer as all-to-all connected lateral input $L$. In setup 2, a sparse connectivity of a maximum of 100 connections is used instead of all-to-all for $L$. In setup 3, additionally to sparse $L$, the label is replaced by a fully connected feedback from the output layer as context $C$, indicating a path towards how multiple hidden layers may be stacked in the future. For comparison, we trained a benchmark network with the same connectivity as setup 1 using standard backpropagation and cross-entropy loss. Additionally, we trained the output layer of a fixed random hidden layer setup using step-function activation.

## 5 RESULTS

**Performance** The performance of all major setups for $N_{\text{hid}} = 100$ is shown in Fig. 3.**B**. The three trivariate infomorphic setups use the same set of optimized goal parameters (see below) for all hidden layer sizes and significantly outperform the random baseline as well as the networks with a bivariate setup for the hidden layer. As shown in Appendix Fig. 3.**C**, Setup 1 matches the performance of backpropagation for up to 100 neurons, and reaches its maximum test accuracy for 500 neurons before

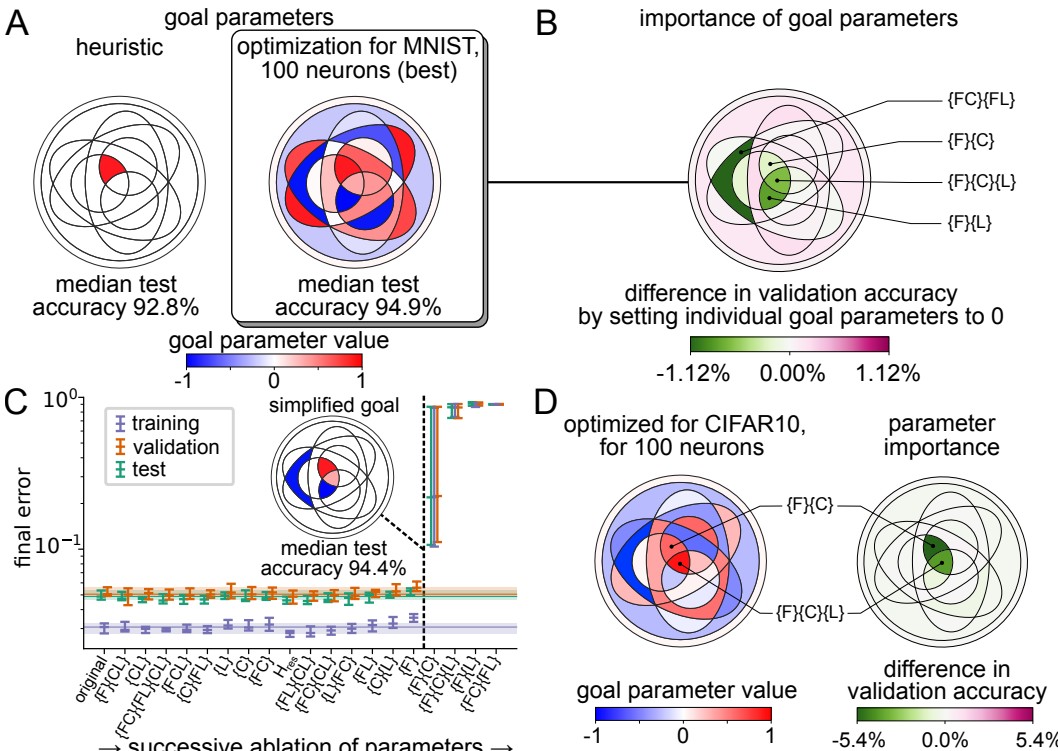

Figure 4: **Given a set of optimized goal parameters, an ablation study allows for the identification and interpretation of the most critical neural subgoals for a given task. A.** The heuristically defined goal function (see Fig. 2.**D**) shows 92.8% test accuracy on MNIST for $N_{hid} = 100$; optimizing the goal function using hyperparameter optimization increases the test accuracy to 94.9%. The optimized goal parameters include the intuitive $\gamma_{\{F\}\{C\}}$, but also additional PID atoms to be maximized or minimized at the same time. **B.** For identifying the most important goal parameters, we performed an ablation study (individually setting $\gamma$ parameters to 0) and measured the change in validation accuracy compared to a network trained with the full goal. **C.** A successive ablation of parameters in order from lowest to highest individual effect identifies four parameters as being crucial for network performance (see Table 3 for their definition). The lines indicate mean values, with the intervals depicting the maximum and minimum of 10 runs. **D.** To test whether more complex image classification tasks require different goal parameters, we perform a separate hyperparameter optimization for setup 1, 100 neurons in CIFAR-10 and reach a median test accuracy of 42.5% (compared to 42.2% using backprop and 41.1% using the goal function optimized for MNIST).

its performance starts decreasing with larger layer sizes. This decrease in accuracy can be attributed to a lack of convergence of the neurons (Appendix Fig. 5), likely arising through the interaction of too many neurons. To alleviate the convergence problem, we performed additional experiments (setup 2) with the number of lateral connections reduced to maximally 100, which leads to better convergence and contributes to the continuous increase of performance for larger hidden layers, reaching a median test accuracy of 97.5% for 2000 neurons, slightly below the 98.0% reached via backpropagation. Finally, experiments with setup 3 provide evidence that the direct label input to the hidden layer can be replaced with feedback from the next layer, while still enabling solid performance especially for large layers. Additionally, we tested the alternate linear activation function as outlined in 4. Here, experiments using the optimized goal function parameters resulted in accuracies between 94.9 % and 95.1 % for 100 neurons, which is on par with the original activation function.

**Goal parameters** The optimized goal function used for training the three main setups at all sizes were obtained by maximizing the validation set accuracy using setup 1 with $N_{\mathrm{hid}} = 100$. We performed a total of six hyperparameter optimizations of 1000 trials, three using TPE and three using CMA-ES samplers, respectively (see Appendix Fig. 6). The best results were achieved using the parameters shown in Fig. 4. The optimized objective functions outperform the intuitive goal function

by including additional PID atoms besides the intuitively derived $\Pi_{\{F\}\{C\}}$. To compare to a more complex image classification task, a single CMA-ES optimization run has been performed for the CIFAR-10 task (Krizhevsky, 2012), with the optimal parameters presented in Fig. 4.**D**.

**Parameter importance**   An analysis of the optimized hyperparameter values (Fig. 4.**B**) shows that only relatively few parameters are of high importance for performance. By setting goal parameters to zero individually (Figure 4.**B**) we find that there are four goal parameters that are critically non-zero for high performance on MNIST. This finding is confirmed by cumulatively setting goal function parameters to zero in the order of the individual drop in performance, which is illustrated in Fig. 4.**C**. Interestingly, we find that CIFAR-10 two of the four critical goal parameters for MNIST seem especially important. As a complementary measure of parameter importance, the results of mean decrease in impurity (Breiman, 2001) can be found in Appendix A.6. The results of a more detailed analysis of parameter importance are presented in Appendix Fig. 9.

**Interpretation**   For the MNIST task, neurons aim for (i) encoding information that is redundant between feedforward and context inputs and thus task relevant, but not already encoded in other neurons (strongly positive $\gamma_{\{F\}\{C\}}$), (ii) avoid encoding information that is not in the context, thus not task-relevant, and already encoded by other neurons (strongly negative $\gamma_{\{F\}\{L\}}$) and (iii) allow redundancies of task relevant information between neurons (slightly positive $\gamma_{\{F\}\{C\}\{L\}}$). Additionally, neurons avoid synergies that require $F$ and either $C$ or $L$ for their recovery ($\gamma_{\{FC\}\{FL\}}$) (see Appendix Table 3 for an overview of the parameter values and interpretations). For the CIFAR-10 task (see Fig. 4.**D**), the two most important parameters $\gamma_{\{F\}\{C\}\{L\}}$ and $\gamma_{\{F\}\{C\}}$ point to a prioritization of redundancy over uniqueness between neurons.

## 6   RELATED WORKS

We are not aware of any other attempts to use PID directly as a constructive tool for designing ANNs with local learning objectives, aside from the bivariate implementation of infomorphic networks (Makkeh et al., 2025). The most directly related work using other local information-theoretic learning objectives is likely the work on coherent infomax by Kay, Phillips and colleagues (Kay and Phillips, 1997). Their idea of defining a neural goal function that ensures only information coherent between the inputs being passed on in the output of a neuron inspired our idea of using PID-based redundancy as a neural goal function. However, as analyzed in Wibral et al. (2017), the lack of a PID at the time of invention of coherent infomax reduced the expressiveness of their goal function, and also prevented an easily interpretable formulation for more than two input classes. Our results suggest that three input classes are crucial to unlock the potential of information theoretic learning, in the sense that a neuron needs to know at least the data to transform ($F$), the relevance of various aspects of these data ($C$), and also what is not yet encoded well by other neurons (via $L$).

Aside from PID proper, there are two prominent principles based on information theory that has been employed to train deep neural networks: information maximization (InfoMax, Linsker, 1988) and information bottleneck (IB, Tishby et al., 2000). The main idea of InfoMax is to learn representations that encode as much information as possible about the task, while IB additionally penalizes information that is irrelevant to the task (Yu et al., 2020). Various frameworks have employed InfoMax and IB where unlike in our framework, information theory is employed at the level of group of neurons (layer). Examples include Deep InfoMax where mutual information of high-level representations and the input is maximized (Hjelm et al., 2019), InfoNCE that maximizes the mutual information of future observations and the current latent representation (Oord et al., 2018), and Variational autoencoders that employ an IB layer (Alemi et al., 2017).

Additionally, local learning rules based on concepts other than information theory have been developed and show potential for scaling to larger, more capable networks (Lillicrap et al., 2020; Richards et al., 2019; Jeon and Kim, 2023; Bredenberg et al., 2024). This includes learning rules based on concepts from contrastive learning (Illing et al., 2021; Ahamed et al., 2023), predictive coding (Mikulasch et al., 2023; Sacramento et al., 2018; Millidge et al., 2022), and many others (Launay et al., 2020; Hinton, 2022; Høier et al., 2023; Lee et al., 2015; Nøkland and Eidnes, 2019; Lässig et al., 2023). Most of these approaches are confined to specific learning paradigms and implementations, while our framework describes local learning goals in an abstract manner, making it applicable to various learning paradigms, datasets and implementations, while still being interpretable.

In terms of abstract per-neuron objectives, recently complementary work has been published, building on concepts from control theory (Moore et al., 2024) and reinforcement learning (Ott, 2020). These efforts are in line with our work, and we believe similar creative, yet principled approaches could provide a basis for deepening the understanding of local information processing.

Beyond the formulation of information theoretic learning rules, there is literature using PID to analyze deep neural network function, such as Tax et al. (2017); Ehrlich et al. (2023); Tokui and Sato (2021); Liang et al. (2023; 2024); Mohamadi et al. (2023). These studies necessarily differ from the perspective on neural function taken here, as they analyze the function of neurons in feedforward networks trained with backprop, where—different to the infomorphic approach—the learning signals are not directly available as inputs to the neural activation function.

## 7 DISCUSSION

In this work we demonstrate how neurons can self-organize to solve classification tasks using local PID-based goal functions. Unlike mechanistic local learning rules or purely global objectives, our framework expresses learning in human interpretable terms. These local objectives are expressed by selecting which output information should be uniquely, redundantly or synergistically determined by the various classes of local inputs to a neuron. We show how to derive a supervised learning objective based on intuitive heuristic reasoning, and upon some optimization of this simple rule demonstrate performance on MNIST that is comparable to the same ANN trained with backpropagation.

By showing which PID atoms need to be optimized to solve a certain goal and assessing the importance of each goal function parameter, infomorphic networks provide a novel insight into the local information processing required at a local level to solve a global goal. Note that this interpretability differs from other notions of interpretability: While other attempts at interpretability focus on how specific tasks are solved, infomorphic neurons show on an abstract level what kind of local information processing is necessary to solve a certain class of problems, e.g., classification. These insights may in the future be used to discern which neurons fail to contribute to the overall task, with the localizability of the $I_\cap^{\text{sx}}$ redundancy measure revealing for which labels problems occur.

**Limitations and Outlook**  We show that the infomorphic local optimization approach reaches a performance comparable to backpropagation in a setup with one fully connected hidden layer. In subsequent work, we will investigate the local information processing and connectivity required for deeper networks. Preliminary experiments show that deeper networks train out of the box using the same goal function parameters and reach the same or slightly higher accuracy compared to the single hidden layer models (see subsection A.10). More work is needed to determine how this accuracy can be improved, for instance by choosing different goal function parameters for the different layers.

The infomorphic neurons presented in this work use a discrete version of PID, which limits the amount of information that individual neurons can convey. While this is similar to the binary output states of "spike" and "no spike" in biological neurons, PID can also be applied to continuous variables and thus could also enable learning in continuous neurons (Ehrlich et al., 2024).

For the study of biological neural networks, infomorphic networks could potentially serve as a powerful constructive modeling approach. While information-theoretic gradients are almost certainly too complex to be directly implemented in biological substrate, biological neurons may still strive for similar objectives through simpler approximations and mechanistic local learning rules. Having a principled and general way to formulate, identify and test local information processing objectives could provide a new perspective on local learning in general. Additionally, subsequent work could aim to establish a formal bridge between local *goals* and known local learning *rules*, reconciling well-established Hebbian-like rules with novel principled approaches.

In conclusion, the formulation of local objective functions in the language of PID directly enables an interpretation of the information processing of neurons. Our work represents an important step towards a principled theory of local learning founded in information theory.

**Code Availability**  The framework and the code to reproduce the results of this work are available under https://github.com/Priesemann-Group/Infomorphic_Networks.

ACKNOWLEDGMENTS

We would like to thank Marcel Graetz, Jonas Dehning, Mark Blümel, Fabian Mikulasch and Lucas Rudelt for their input and fruitful discussions about this topic. We would also like to thank Johannes Zierenberg, Sebastian Mohr, Darius Burr and the rest of the Priesemann and the Wibral Group for their valuable comments and feedback on this work. A.S., V.N., A.E. and V.P. were funded via the MBExC by the Deutsche Forschungsgemeinschaft (DFG, German Research Foundation) under Germany's Excellence Strategy-EXC 2067/1-390729940. A.S., V.N., A.E., V.P., and M.W., were supported and funded by the DFG – GRK2906 – project number 502807174. V.N. was partly supported by the Else Kröner Fresenius Foundation via the Else Kröner Fresenius Center for Optogenetic Therapies. D.E. and M.W. were supported by a funding from the Ministry for Science and Education of Lower Saxony and the Volkswagen Foundation through the "Niedersächsisches Vorab" under the program "Big Data in den Lebenswissenschaften" – project "Deep learning techniques for association studies of transcriptome and systems dynamics in tissue morphogenesis". A.M. and M.W. are employed at the Campus Institute for Dynamics of Biological Networks (CIDBN) funded by the Volkswagenstiftung. A.E., V.P. and M.W. received funding from the DFG via the SFB 1528 "Cognition of Interaction" - project-ID 454648639. A.E. was supported from the European Research Council (ERC) under the European Union's Horizon Europe research and innovation programme (Grant agreement No. 101041669). M.W. was supported by the flagship science initiative of the European Commission's Future and Emerging Technologies program under the Human Brain project, HBP-SP3.1-SGA1-T3.6.1.

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

## A APPENDIX

### A.1 PSEUDOCODE FOR TRAINING ALGORITHM (SETUP 1)

In this section, we provide pseudocode detailing the procedure to train infomorphic networks with dense lateral connections and label context (i.e., Setup 1 in Figure 3). The code is started from the main function TrainModel, which invokes TrainNeuronTrivariate and TrainNeuronBivariate to update hidden layer and output layer weights, respectively. These functions in turn make use of the auxiliary functions ComputeIsxRedundancies, which computes generalized redundancies according to the analytical definition by Makkeh et al. (2021), and ComputePIDAtoms, which performs a Moebius inversion to compute the PID atoms from the generalized redundancies (Williams and Beer, 2010). Note that since the linear operation performed by ComputePIDAtoms is fixed, it can be efficiently implemented as a multiplication with a precomputed matrix.

Note that, as originally proposed by Kay (1994) and utilized by Makkeh et al. (2025), gradients for the incoming weights are only computed with respect to the conditional probability $p(y|f, c, l)$ (or $p(y|f, c)$) of the output $Y$, since the empirical probability mass functions $p(f, c, l)$ in Algorithm 2 and $p(f, c)$ in Algorithm 3 are not differentiable. While we currently have no strong a priori argument why the empirical probability mass functions can be assumed constant, the learning of PID over training and the good task performance when using these gradients in experiments provides an ex post justification for this procedure.

---

**Function 1:** TrainModel

**Input:** data, num_epochs
**Output:** trained model

1   INITIALIZE model;
2   **foreach** epoch in range(num_epochs) **do**
3     **foreach** (batch_samples, batch_labels) in data **do**
4       INITIALIZE hidden_state, output_state to zero;
5       hidden_state $\leftarrow$ forward_hidden_layer(f=batch_samples, c=batch_labels, l=hidden_state);
6       hidden_state $\leftarrow$ forward_hidden_layer(f=batch_samples, c=batch_labels, l=hidden_state);
7       output_state $\leftarrow$ forward_output_layer(f=hidden_state, c=batch_labels) ;
8       **foreach** neuron in hidden_layer **do**
9         TrainNeuronTrivariate(y=hidden_state[neuron], f=batch_samples, c=batch_labels, l=last_outputs[other neurons]);
10       **foreach** neuron in output_layer **do**
11         TrainNeuronBivariate(y=output_state[neuron], f=hidden_state, c=batch_labels);
12   **return** trained model

---

---

**Function 2:** TrainNeuronTrivariate

**Input:** y, f, c, l

1 BIN continuous values f, c, l to 20 equally sized bins;
2 COUNT occurrences of tuples (f, c, l) in batch;
3 COMPUTE empirical probability masses p(f, c, l);
4 EVALUATE conditional probabilities p(y|f, c, l) from the neurons;
5 CONSTRUCT full joint probability mass function $p(y, f, c, l) = p(f, c, l)p(y|f, c, l)$;
6 isx_redundancies ← ComputeIsxRedundancies(p(y, f, c, l));
7 pid_atoms ← ComputePIDAtoms(isx_redundancies);
8 goal ← scalar_product(goal_params, pid_atoms);
9 PERFORM autograd to maximize goal;
10 UPDATE neuron weights;

---

**Function 3:** TrainNeuronBivariate

**Input:** y, f, c

1 BIN continuous values f, c to 20 equally sized bins;
2 COUNT occurrences of tuples (f, c) in batch;
3 COMPUTE empirical probability masses p(f, c);
4 EVALUATE conditional probabilities p(y|f, c) from the neurons;
5 CONSTRUCT full joint probability mass function $p(y, f, c) = p(f, c)p(y|f, c)$
  isx_redundancies ← ComputeIsxRedundancies(p(y, f, c));
6 pid_atoms ← ComputePIDAtoms(isx_redundancies);
7 goal ← scalar_product(goal_params, pid_atoms);
8 PERFORM autograd to maximize goal;
9 UPDATE neuron weights;

---

**Function 4:** ComputeIsxRedundancies

**Input:** Joint probability mass function p(y, f, c, l)

1 **foreach** antichain $\alpha$ **do**
2 $\quad$ COMPUTE conditional probability mass functions $\mathbb{P}(Y = y| \bigvee_{\boldsymbol{a} \in \alpha} \bigwedge_{i \in \boldsymbol{a}} S_i = s_i)$;
3 $\quad$ COMPUTE marginal probability mass function $\mathbb{P}(Y = y)$;
4 $\quad$ $I_\cap^{\mathrm{sx}}(Y : S_\alpha) \leftarrow \sum_{y,f,c,l} \mathbb{P}(Y = y, F = f, C = c, L = l) \log_2 \frac{\mathbb{P}(Y=y| \bigvee_{\boldsymbol{a} \in \alpha} \bigwedge_{i \in \boldsymbol{a}} S_i = s_i)}{\mathbb{P}(Y=y)}$;
5 **return** $I_\cap^{\mathrm{sx}}(Y : S_\alpha)$ for all antichains $\alpha$

---

**Function 5:** ComputePIDAtoms

**Input:** $I_\cap^{\mathrm{sx}}(Y : S_\alpha)$ for all antichains $\alpha$

1 COMPUTE pid atoms $\Pi_\beta$ by inverting the linear system of equations $I_\cap^{\mathrm{sx}}(Y : S_\alpha) = \sum_{\beta \preceq \alpha} \Pi_\beta$
  where $\beta \preceq \alpha \Leftrightarrow \forall \boldsymbol{a} \in \alpha \, \exists \boldsymbol{b} \in \beta : \boldsymbol{a} \subseteq \boldsymbol{b}$;
2 **return** $\Pi_\beta$ for all antichains $\beta$

---

## A.2 DEFINITION OF SHARED-EXCLUSION REDUNDANCY

In this section, we briefly motivate and explain the shared-exclusion redundancy measure $I_\cap^{\text{sx}}$ introduced by Makkeh et al. (2021). The mutual information $I(T : S_1, S_2)$ between a target variable $T$ and two source variables $S_1$ and $S_2$ can be interpreted in a Bayesian sense as an average measure for how the prior belief of the target event $T = t$, needs to be updated in light of the event of observing *both* source events $S_1 = s_1$ *and* $S_2 = s_2$ simultaneously:

$$I(T : S_1, S_2) = \sum_{t, s_1, s_2} \mathbb{P}(T = t, S_1 = s_1, S_2 = s_2) \log_2 \frac{\mathbb{P}(T = t | S_1 = s_1 \wedge S_2 = s_2)}{\mathbb{P}(T = t)}.$$

Makkeh et al. (2021) build on this logic and define redundancy as an average measure for how the beliefs about the target event $T = t$ need to be updated if instead it is only known that $S_1 = s_1$ *or* $S_2 = s_2$ have occurred:

$$I_\cap^{\text{sx}}(T : S_1, S_2) = \sum_{t, s_1, s_2} \mathbb{P}(T = t, S_1 = s_1, S_2 = s_2) \log_2 \frac{\mathbb{P}(T = t | S_1 = s_1 \vee S_2 = s_2)}{\mathbb{P}(T = t)}.$$

For more than two source variables $s_i$ (where i is an index enumerating the set of source variables), the term to condition on becomes a disjunction between conjunctions of the form $\bigvee_{\boldsymbol{a} \in \alpha} \bigwedge_{i \in \boldsymbol{a}} S_i = s_i$ for the redundancy associated with the antichain $\alpha$. The atoms $\Pi$ can then be computed from these redundancies via a Moebius inversion, as laid out in detail in Gutknecht et al. (2021).

The definition is symmetric with respect to permutation of the sources, fulfills a target chain rule and is differentiable with respect to the underlying probability distribution (Makkeh et al., 2021), which makes it a suitable definition for optimizing objective functions.

## A.3 COMPUTE RESOURCES

For the figures in this paper, we performed a total of seven hyperparameter optimizations: Three TPE optimizations and three CMA-ES optimizations of MNIST objective function parameter and a single CMA-ES optimization for the CIFAR-10 task. Each of these optimizations took $\leq 12$ hours to compute on a HPC cluster node with 32 cpu cores and two NVIDIA A100 GPUs with $40\,\text{GB}$ of VRAM each.

Including evaluation runs and earlier computations not shown in the results, we estimate to have utilized a total of 200 hours of compute time on a compute node equivalent to the one described earlier.

## A.4 IMPLEMENTATION DETAILS AND MODEL PARAMETERS

We implemented infomorphic networks as a flexible and efficient python package using pytorch (Paszke et al., 2019) for automatic differentiation of the local goal functions. Furthermore, the optuna (Akiba et al., 2019) package was used to compute the TPE and CMA-ES hyperparameter optimizations as well as the computation of the parameter importance via the mean decrease in impurity.

In this paper, we use the MNIST (LeCun et al., 2015) and CIFAR-10 (Krizhevsky, 2012) datasets. The MNIST dataset consists of 70,000 grayscale images of handwritten digits, each sized 28x28 pixels. The dataset is split into a training set of 60,000 samples and a test set of 10,000 samples. The CIFAR-10 dataset consists of 60,000 images with three color channels and a resolution of 32x32 pixels. Of the 60,000 images, 50,000 are in the training set and 10,000 are in the test set. For each of our training runs with either dataset, $20\%$ of the training set samples are withheld randomly to be used for validation.

The framework and the code to reproduce the results of this work are available under https://github.com/Priesemann-Group/Infomorphic_Networks.

The parameters used to train the models shown in Fig. 3 are listed in Table 1. The backpropagation model was trained to optimize the cross-entropy loss between the prediction and the true label.

Table 1: Model and training parameters for the models shown in Fig. 3. *The effect of different batch sizes is analyzed in Fig. 10. **The effect of the number of bins is analyzed in Fig. 11

| Parameter | Backprop Model | Hidden Layer | Output Layer |
|---|---|---|---|
| $N_{\mathrm{Epochs}}$ | 100 | 100 | 100 |
| $N_{\mathrm{Batch}}$ | 1024 | 1024* | 1024* |
| Optimizer | Adam | Adam | Adam |
| Learning rate $\eta$ | 0.001 | 0.002 | 0.003 |
| Weight decay | 0.0 | 0.00035 | 0.00015 |
| Number of bins per dim. | - | 20** | 20** |
| Binning ranges | - | (-20,20) | adaptive |
| Objective function | cross-entropy | trivariate $\gamma$ | $\gamma = (-0.2, 0.1, 1.0, 0.1, 0.0)^T$ |

### A.5 MODEL DYNAMICS DURING TRAINING

As a measure for the dynamics of the neurons during training, we introduce the self-cosine distance of the feedforward receptive field as

$$D_c^{(t)} = 1 - \frac{\boldsymbol{w}_F^{(t-1)} \cdot \boldsymbol{w}_F^{(t)}}{\|\boldsymbol{w}_F^{(t-1)}\| \|\boldsymbol{w}_F^{(t)}\|} \tag{4}$$

where $\boldsymbol{w}_F$ corresponds to feedforward weights of a single neuron of the hidden layer. The median value of $D_c^{(t)}$ for the hidden neurons of a trivariate model are shown in Fig. 5. While the self-cosine distance consistently increases with layer size in the dense setups, it does not increase in the sparse setup after a layer size of 100 neurons, which is also the number of lateral connections in all of the larger sparse layers.

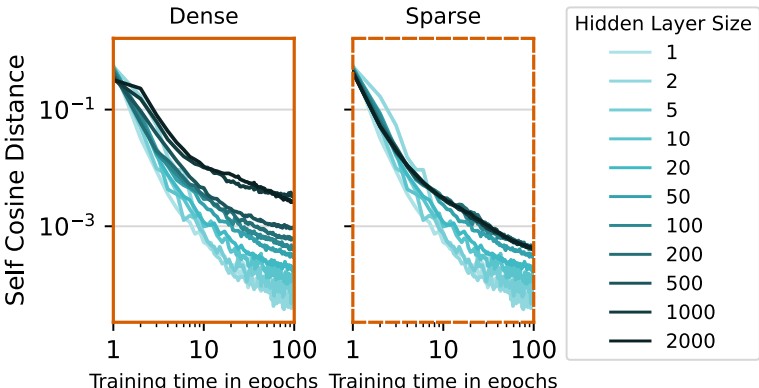

Figure 5: The median self-cosine distance of trivariate neurons for different layer sizes during the course of training in a dense (left) and a sparse (right) lateral connected setup.

### A.6 OPTIMIZED OBJECTIVE FUNCTIONS FOR THE TRIVARIATE NEURONS

To support the trivariate objective function that was obtained using hyperparameter optimization, we repeated the optimizations three times for both, the TPE sampler and the CMA-ES sampler, with parameter values being limited to the interval $\gamma_i \in [-1, 1]$. The resulting objective functions are illustrated in Fig. 6. While the objective functions that were optimized with the TPE sampler included all $\gamma$ parameters, the optimization with the CMA-ES sampler was only performed for the parameters that describe PID-atoms while the parameter for the residual entropy was set to 0. This was done because we observed a strong correlation between the parameter for the unique information of the feedforward input and the residual entropy which both had to be small. In the figure, the respective

values are set 0. In the figure, we also illustrate the mean decrease impurity score which is a measure for the importance of a parameter.

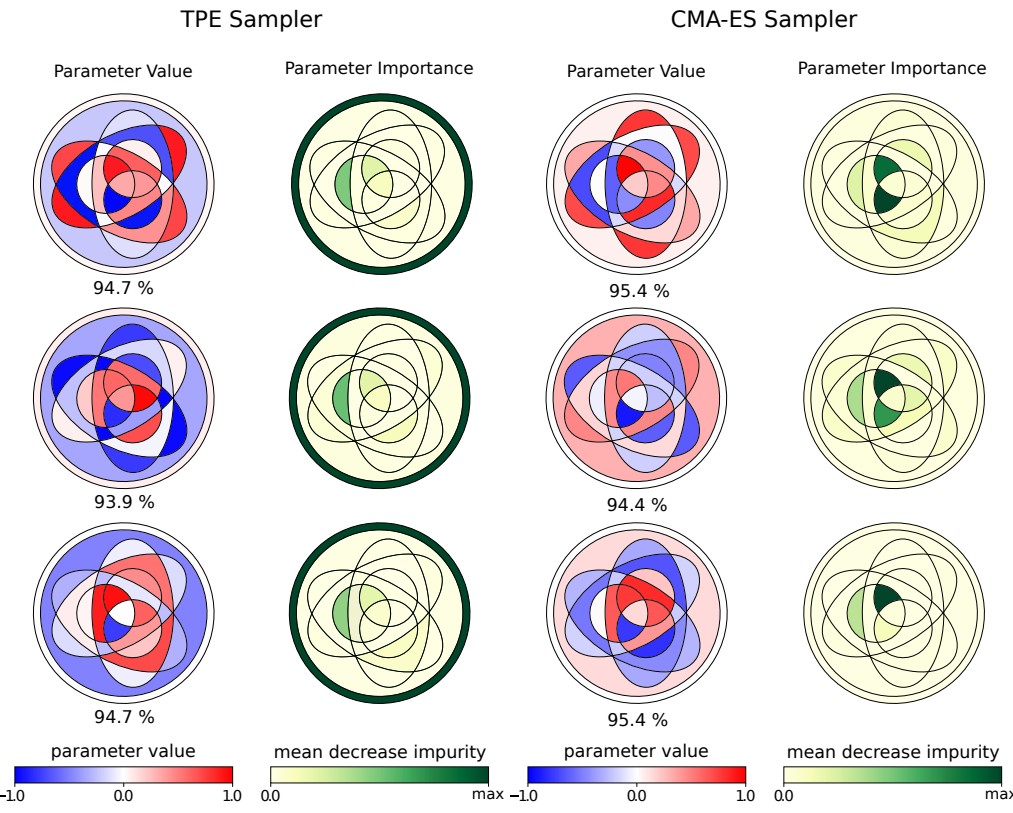

Figure 6: The three objective functions shown in the first column were obtained by utilizing the TPE sampler while the objective functions in the third column were obtained with the CMA-ES sampler. The performance illustrated below each of the objective functions corresponds to the maximum validation accuracy during optimization. The column next to the objective functions shows the corresponding mean decrease impurity score of the corresponding goal parameter as a measure for the importance of the parameters. While the objective functions obtained with CMA-ES have a higher maximum validation accuracy than the goals obtained with the TPE, we observed that the first objective function obtained with the TPE sampler (top-left) outperformed the others in terms of median validation accuracy for larger layer sizes, which is why we used this objective function for the main results.

## A.7 TRIVARIATE CONSISTENCY EQUATIONS

For arbitrary numbers of sources, the PID consistency equations can be written as

$$I(Y : \boldsymbol{a}) = \sum_{\alpha \subseteq a} \Pi_\alpha. \tag{5}$$

In particular, the seven consistency equations for three input variables $F$, $C$ and $L$ can be written as

$$I(Y : F) = \Pi_{\{F\}} + \Pi_{\{F\}\{C\}} + \Pi_{\{F\}\{L\}} + \Pi_{\{F\}\{CL\}} + \Pi_{\{F\}\{C\}\{L\}}$$

$$I(Y : C) = \Pi_{\{C\}} + \Pi_{\{F\}\{C\}} + \Pi_{\{C\}\{L\}} + \Pi_{\{C\}\{FL\}} + \Pi_{\{F\}\{C\}\{L\}}$$

$$I(Y : L) = \Pi_{\{L\}} + \Pi_{\{F\}\{L\}} + \Pi_{\{C\}\{L\}} + \Pi_{\{L\}\{FC\}} + \Pi_{\{F\}\{C\}\{L\}}$$

$$I(Y : F, C) = \Pi_{\{F\}} + \Pi_{\{C\}} + \Pi_{\{F\}\{C\}} + \Pi_{\{F\}\{L\}} + \Pi_{\{C\}\{L\}} + \Pi_{\{F\}\{CL\}}$$
$$+ \Pi_{\{C\}\{FL\}} + \Pi_{\{L\}\{FC\}} + \Pi_{\{FC\}\{FL\}\{CL\}} + \Pi_{\{FC\}\{FL\}} + \Pi_{\{FC\}\{CL\}}$$
$$+ \Pi_{\{F\}\{C\}\{L\}} + \Pi_{\{FC\}}$$

$$I(Y : F, L) = \Pi_{\{F\}} + \Pi_{\{L\}} + \Pi_{\{F\}\{C\}} + \Pi_{\{F\}\{L\}} + \Pi_{\{C\}\{L\}} + \Pi_{\{F\}\{CL\}}$$
$$+ \Pi_{\{C\}\{FL\}} + \Pi_{\{L\}\{FC\}} + \Pi_{\{FC\}\{FL\}\{CL\}} + \Pi_{\{FC\}\{FL\}} + \Pi_{\{FL\}\{CL\}} \quad (6)$$
$$+ \Pi_{\{F\}\{C\}\{L\}} + \Pi_{\{FL\}}$$

$$I(Y : C, L) = \Pi_{\{C\}} + \Pi_{\{L\}} + \Pi_{\{F\}\{C\}} + \Pi_{\{F\}\{L\}} + \Pi_{\{C\}\{L\}} + \Pi_{\{F\}\{CL\}}$$
$$+ \Pi_{\{C\}\{FL\}} + \Pi_{\{L\}\{FC\}} + \Pi_{\{FC\}\{FL\}\{CL\}} + \Pi_{\{FC\}\{CL\}} + \Pi_{\{FL\}\{CL\}}$$
$$+ \Pi_{\{F\}\{C\}\{L\}} + \Pi_{\{CL\}}$$

$$I(Y : F, C, L) = \Pi_{\{F\}} + \Pi_{\{C\}} + \Pi_{\{L\}} + \Pi_{\{F\}\{C\}} + \Pi_{\{F\}\{L\}} + \Pi_{\{C\}\{L\}}$$
$$+ \Pi_{\{F\}\{CL\}} + \Pi_{\{C\}\{FL\}} + \Pi_{\{L\}\{FC\}} + \Pi_{\{FC\}\{FL\}\{CL\}} + \Pi_{\{FC\}\{CL\}}$$
$$+ \Pi_{\{FL\}\{CL\}} + \Pi_{\{FC\}\{FL\}} + \Pi_{\{F\}\{C\}\{L\}} + \Pi_{\{FC\}} + \Pi_{\{FL\}} + \Pi_{\{FL\}}$$
$$+ \Pi_{\{FCL\}}.$$

### A.8 THE ATOMS OF THE PARTIAL INFORMATION DECOMPOSITION FOR THREE SOURCE VARIABLES AND THEIR INTERPRETATION

To improve the clarity and intuition for the atoms of the partial information decomposition with three source variables, Fig. 7 shows a larger version of decomposition diagram already shown in Fig. 2.**C**. As a guide for the reader, the meaning of the different information atoms are listed in Table 2. Additionally, Table 3 explicitly lists the goal parameter values found via optimization together with their interpretation.

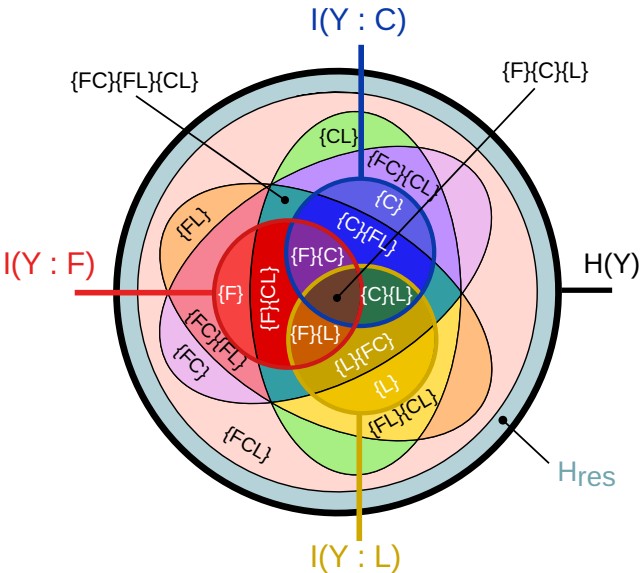

Figure 7: Larger version of decomposition diagram already shown in Fig. 2.

Table 2: A list of all atoms that are part of the Partial Information Decomposition with three source variables and their meaning.

| Atom ($\Pi_{\text{Antichain}}$) | Meaning |
|---|---|
| $\Pi_{\{F\}\{C\}\{L\}}$ | information that is redundant in all of the three sources |
| $\Pi_{\{F\}\{C\}}$ | redundant information between feedforward and context input |
| $\Pi_{\{F\}\{L\}}$ | redundant information between feedforward and lateral input |
| $\Pi_{\{C\}\{L\}}$ | redundant information between context and lateral input |
| $\Pi_{\{F\}\{CL\}}$ | information provided by both, the feedforward input and the synergy between the context and the lateral input |
| $\Pi_{\{C\}\{FL\}}$ | information provided by both, the context input and the synergy between the feedforward and the lateral input |
| $\Pi_{\{L\}\{FC\}}$ | information provided by both, the lateral input and the synergy between the feedforward and the context input |
| $\Pi_{\{F\}}$ | unique information provided by the feedforward input |
| $\Pi_{\{C\}}$ | unique information provided by the context input |
| $\Pi_{\{L\}}$ | unique information provided by the lateral input |
| $\Pi_{\{FC\}\{FL\}\{CL\}}$ | information redundantly provided by each of the pairwise synergies |
| $\Pi_{\{FC\}\{FL\}}$ | information provided by both, the synergy between the feedforward and the context input and the synergy between the feedforward and the lateral input |
| $\Pi_{\{FC\}\{CL\}}$ | information provided by both, the synergy between the feedforward and the context input and the synergy between the context and the lateral input |
| $\Pi_{\{FL\}\{CL\}}$ | information provided by both, the synergy between the feedforward and the lateral input and the synergy between the context and the lateral input |
| $\Pi_{\{FC\}}$ | synergy between the feedforward input and the context input |
| $\Pi_{\{FL\}}$ | synergy between the feedforward input and the lateral input |
| $\Pi_{\{CL\}}$ | synergy between the context input and the lateral input |
| $\Pi_{\{FCL\}}$ | information encoded by all input sources synergistically |

Table 3: The four most important non-zero goal parameters found through optimization on MNIST and ablation lead to interpretable requirements on each neuron's local information processing. Note that the optimization was performed with parameter values being limited to the interval $\gamma_i \in [-1, 1]$.

| parameter | definition | value | interpretation of local information processing goal |
|---|---|---|---|
| $\gamma_{\{F\}\{C\}}$ | redundancy between $F$ and $C$ | 0.98 | maximize information that is provided both by feedforward AND context but not by other neurons—and thus relevant for the task |
| $\gamma_{\{F\}\{L\}}$ | redundancy between $F$ and $L$ | -0.99 | minimize information that is redundant with other neurons, but NOT provided by the context—and thus not relevant for the task |
| $\gamma_{\{F\}\{C\}\{L\}}$ | redundancy between $F$, $C$ and $L$ | 0.33 | moderately maximize information that is redundant with other neurons AND the context—thus relevant, but already encoded |
| $\gamma_{\{FC\}\{FL\}}$ | redundancy between synergies requiring $F$ | -0.97 | minimize synergistic information that requires $F$ and either $C$ or $L$ to be recovered |

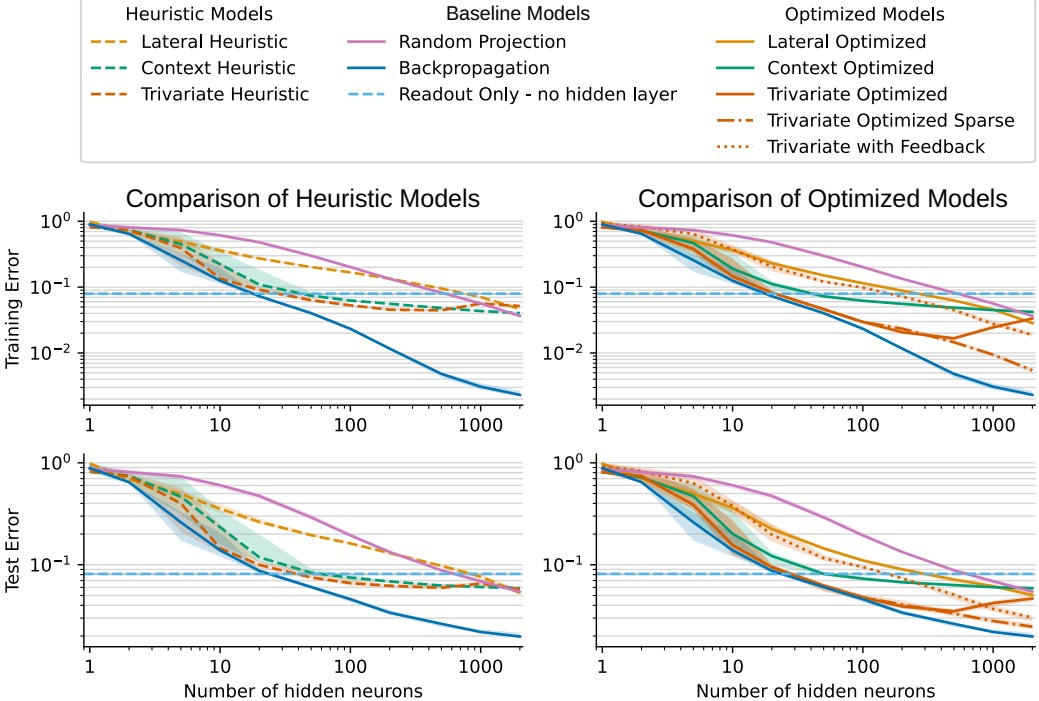

Figure 8: An overview over the training and test performances of the heuristic (left) and optimized (right) models with different model structures, neuron models and objective functions as well as the performances of some baseline models. In the bivariate models, only one of the non driving inputs (context or lateral) and the neurons optimized the heuristic objective functions illustrated in Fig. 2.**D**. The corresponding optimized objectives were optimized with the CMA-ES sampler similar to the trivariate objective function.

## A.9    RECURRENCE OF THE HIDDEN LAYER

To perform an information decomposition which includes the information of the lateral activity, we include lateral connections within the hidden layer as illustrated in Fig. 3.**A**. As described in Appendix A.1, we only perform two iterations within the hidden layer i.e.,one iteration to compute the activity of the neurons within the layer and a second iteration to include this activity in the final output of the neurons. Therefore, the lateral activity contributes to the neurons output. Due to the stochasticity of the neurons, the lateral input will never fully converge. The continuous output probabilities converge already after the second single iteration with only minor changes in the output probabilities of the order of $10^{-4}$. This is due to the small influence of the lateral input on the output which is limited to $0.1\,\sigma(2FL)$ i.e., a maximum change of $\pm 0.1$ in the activation.

## A.10    DEEP INFOMORPHIC NETWORKS

A major factor in the success in many neural network architectures is their layered structure, allowing the network to process the relevant information in successive steps to make them more accessible to the final output layer. For this reason, scaling infomorphic networks to multiple hidden layers is the canonical next step for making them applicable to more complex tasks.

In preliminary experiments we found that networks with more hidden layers work out-of-the-box (without any further optimization of the goal or other hyperparameters) and perform equally well or slightly better than comparable shallower networks. We provide two examples: First, networks with layer sizes 800-200-10 and 600-200-200-10 (each layer receiving the label as context) reach a mean final test accuracy of 97.2 % and 97.1 % on MNIST. This matches the performance of architecture 1000-10 (setup 2) with a slightly smaller model complexity (same number of neurons,

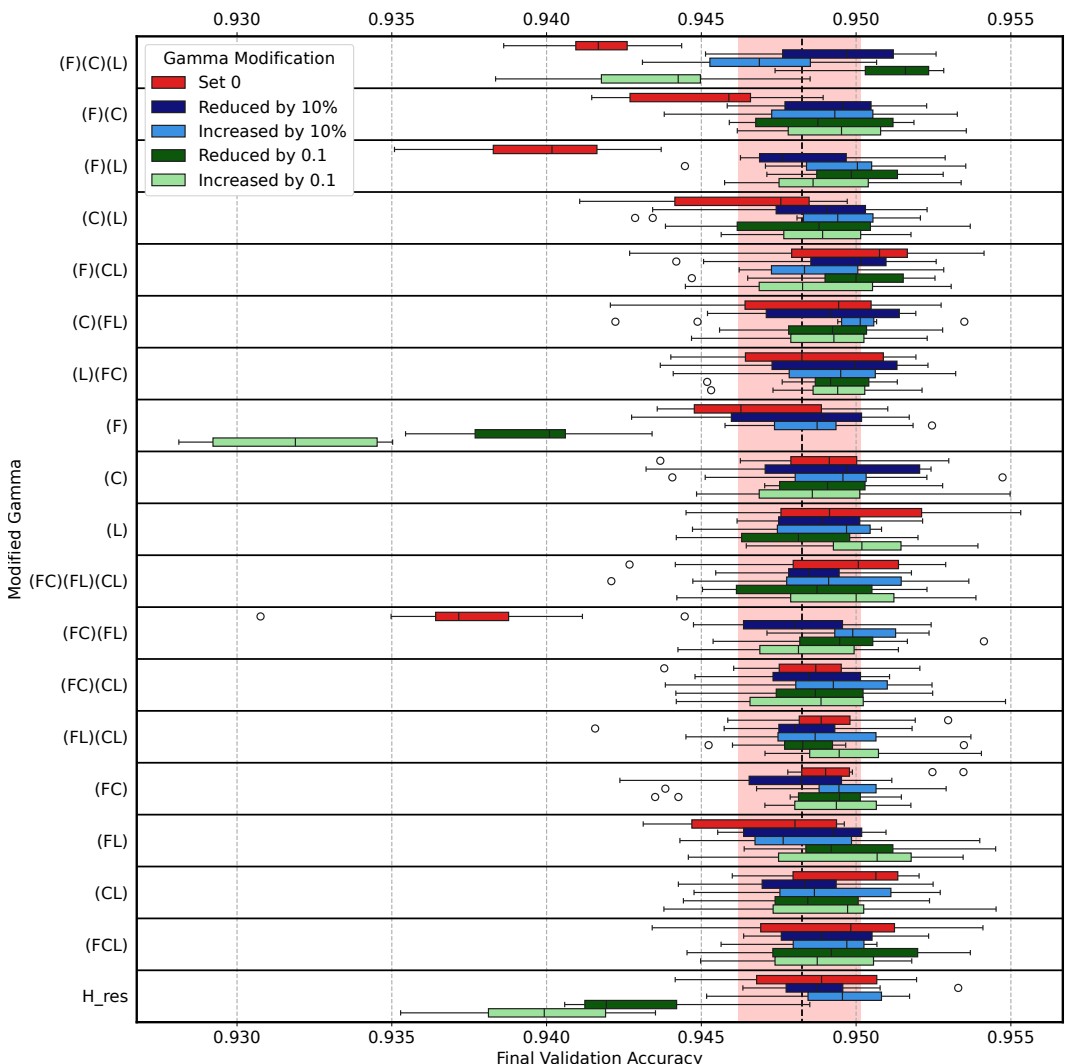

Figure 9: To estimate the importance of the different parameters of the trivariate objective function, we modified individual parameters of the trivariate objective function. The performance change induced by setting the parameters to 0 was already shown in Fig. 4.**B** and was used for the results shown in Fig. 4.**C**. Additionally, we changed the individual goal parameter relative to the parameter strength ($\pm 10\%$), but also in an absolute manner ($\pm 0.1$) to get a better intuition about the importance of the fine tuning of the parameters.

fewer parameters). Second, an architecture of 1600-400-10 (each layer receiving label and feedback from the output layer as context) reaches a mean final test accuracy of 97.7 %, outperforming the performance of 2000-10 (setup 2) with 97.5 %. These results provide evidence that stacking multiple layers works in principle.

In subsequent work, we will optimize the goal function parameters for a feedback setup to evaluate the efficacy of this approach and uncover differences to the setup with the label as context signal. Furthermore, we will conduct an investigation into how the optimal goal parameters differ between individual hidden layers of a deeper network.

As shown in figure 2.**D** we now created the goal function for the trivariate neurons by maximizing the atom that corresponds to the overlap of the two bivariate atoms. This atom corresponds to the redundant information between the feedforward and the contextual input, that is uniquely provided by the two sources and not by the lateral input, resulting in the heuristic goal function of the form $G_{\text{triv}} = \Pi_{\{F\}\{C\}}$. It is important to note, that even if this goal looks identical to the bivariate goal that

maximizes the redundant information between F and C, it is different as the meaning of the atoms depends on the number of sources despite the identical notation.

## A.11 BATCH SIZE AND NUMBER OF BINS

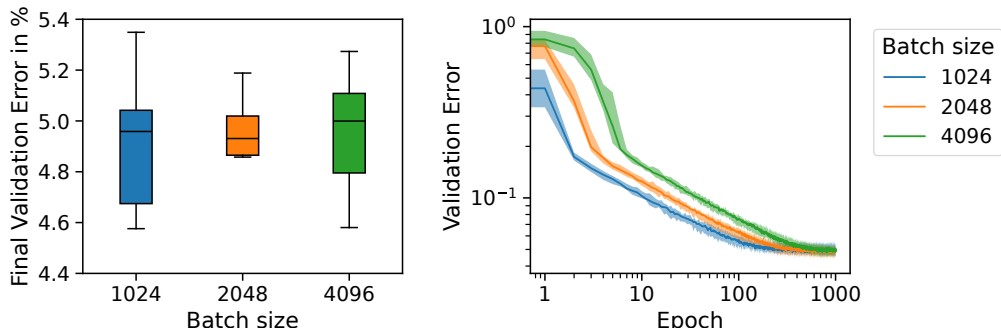

Figure 10: To evaluate whether a batch size of 1024 is sufficient for estimating the probability distributions required for information decomposition, we conducted additional experiments with the trivariate model. These experiments utilized the optimized goal function, and a hidden layer size of $N_{\text{hid}} = 100$, while varying the batch size. Each configuration was run in 10 different initializations and trained for 1000 epochs to ensure convergence, as demonstrated in the right figure. The left figure reveals that performance remains consistent across different batch sizes. However, models with smaller batch sizes converge faster, reducing the computational cost of optimization. Consequently, a batch size of 1024 appears adequate for the requirements of this study.

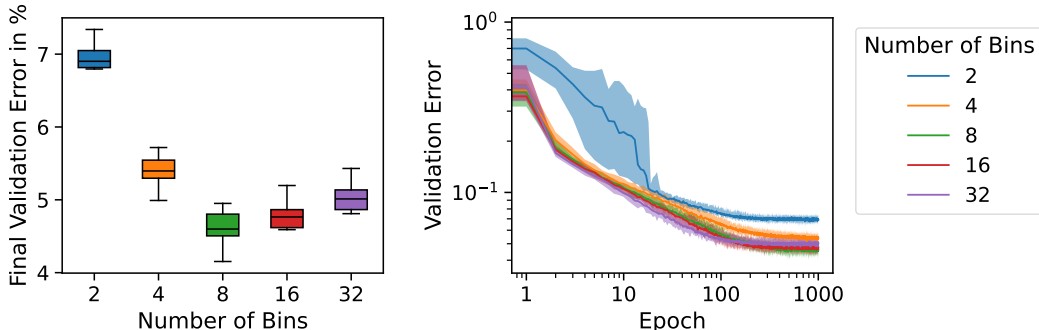

Figure 11: Building on our analysis of the dependence of training on batch size, we conducted similar experiments, this time varying the number of bins used to estimate probability distributions. Theoretically, using too few bins results in greater information loss, as more distinct samples are grouped into the same bin. Conversely, too many bins make it challenging to reliably estimate information-theoretic quantities. As shown in the figure, a high number of bins accelerates convergence, while too few bins introduce noise into the training process. Consistent with theoretical expectations, the optimal number of bins falls within a balanced range.

