# OpenReview forum: "What should a neuron aim for? Designing local objective functions based on information theory"
_ICLR.cc/2025/Conference — ICLR 2025 Oral_

### Official Review · Reviewer_wfkU · 2024-10-31

**Soundness:** 2
**Presentation:** 2
**Contribution:** 3
**Rating:** 6
**Confidence:** 3

**Summary:**

This paper introduces trivariate infomorphic neuron model to develop an interpretable and biologically-inspired local objective function for training artificial neural networks. This abstract neuron model is based on Partial Information Decomposition (PID) framework, which decomposes information into unique, redundant, and synergistic components (atoms). Local objective (goal) function is constructed as a linear combination of these PID information atoms. Neural network models with one hidden layer and various configurations (e.g. sparse lateral connectivity, different context signal) are trained with the proposed loss function on MNIST and CIFAR10 classification tasks, demonstrating promising results.

**Strengths:**

* The problem is well-stated in the introduction.

* Section 3 and Section 4 effectively introduce the PID framework and explain how it is used to develop local objective functions for training neural networks.

* The shared code is well-structured, which aids the reproducibility of the work.

* MNIST results seem promising.

**Weaknesses:**

The paper has a few areas that could benefit from further clarification or expansion. Please see the following points and the Questions section for more details.

*  If I did not overlook it, the consistency equations for the trivariate infomorphic neuron model are not included in the paper.

* The discussion of weight updates is limited. It is mentioned that either autograd or the analytical formulation in (Makkeh et al. 2023) can be used. However, the analytical approach in (Makkeh et al. 2023) applies to  bivariate informorphic neuron model. Although the weight update derivation is potentially a similar approach, the absence of an explicit derivation raises a concern. Without this, I am uncertain if the resulting learning is biologically plausible.

* It is mentioned that (line 308) the $\gamma$ parameters can be derived from intuitive notions. However, unless I overlooked something, the explanation for determining these parameters is not provided in detail beyond a brief note in the caption of Figure 2.D.

* I am concerned that plug-in estimation of the information atoms based on empirical probability mass functions may face challenges in high-dimensional settings due to the difficulty of density estimation. While MNIST may be manageable despite being high-dimensional, this could be an issue in more complex settings.

* Lines 511-515 mention that deeper networks were trained and achieved comparable or better accuracy results; however, these results are not shown. The paper only includes experiments with a single hidden layer.

* The algorithm in Function 2 (Appendix A.1) requires quantities labeled 'isx_redundancies' and 'pid_atoms' (lines 732 and 733), but it is not specified how these quantities are computed.

* According to Appendix A.3, the proposed method requires a lot of computational power even for MNIST. This raises concerns about its scalability to more complex datasets.

* I think the abbreviation IM in Table 1 likely stands for "infomorphic networks," but this is not clarified in the paper.

**Minor Comments:**

* In line 907, the word 'function' is repeated.

* I think Figure 9 is neither referenced in the text nor is it fully interpreted beyond the caption. Given its complexity, more discussion could make it easier to understand.

**Questions:**

* Is the activation function in line 283 commonly used? How the $\alpha_i$ and $\beta_i$ (for $i = 1, 2$) values are determined for this activation function?

* Regarding lines 293-297, which mention that lateral connections introduce recurrence, how did you determine that presenting the same input twice is sufficient? For more complex datasets, more iterations may be needed. I am curious if the output converges after two presentations.

* Is there any particular heuristic or rationale for using 20 equally sized bins in the algorithms in Appendix A.1?

---

> ### Author Response · Authors · 2024-11-21
>
> > If I did not overlook it, the consistency equations for the trivariate infomorphic neuron model are not included in the paper.
>
> For completeness, we added the consistency equations for trivariate Partial Information Decomposition as an additional appendix section.
>
> > The discussion of weight updates is limited. It is mentioned that either autograd or the analytical formulation in (Makkeh et al. 2023) can be used. However, the analytical approach in (Makkeh et al. 2023) applies to bivariate informorphic neuron model. Although the weight update derivation is potentially a similar approach, the absence of an explicit derivation raises a concern. Without this, I am uncertain if the resulting learning is biologically plausible.
>
> While analytical gradients can in principle be derived for the trivariate case, we have so far not seen a direct benefit for doing so over using autograd. Please note that the autograd approach results in gradients mathematically equivalent to the explicit analytical formulation, which we have also confirmed empirically for the bivariate case. Since therefore the weight updates take the same local inputs and produce the same outputs, the biological plausibility remains unaffected. As outlined in the discussion section, we do not expect biological systems to implement PID learning goals directly, but they nevertheless may have mechanistic learning rules which implicitly optimize for a similar information-theoretic goal.
>
> > It is mentioned that (line 308) the  parameters can be derived from intuitive notions. However, unless I overlooked something, the explanation for determining these parameters is not provided in detail beyond a brief note in the caption of Figure 2.D.
>
> As outlined in the last paragraph of Section 3, the intuition behind the trivariate goal function stems from the combination of two intuitive requirements for the neuron's goal: Firstly, a neuron should encode in its output information which is coherently encoded in both the input to the network and the label. Secondly, the a neuron should provide information in its output that is unique with respect to what the other neurons already encode. Combining these notions together, the neuron should intuitively encode that part of the information which is redundant between the feedforward and context inputs while simultaneously being unique with respect to the other neurons of the same layer. We highlighted this section better in our manuscript.
>
> > I am concerned that plug-in estimation of the information atoms based on empirical probability mass functions may face challenges in high-dimensional settings due to the difficulty of density estimation. While MNIST may be manageable despite being high-dimensional, this could be an issue in more complex settings.
>
> Please refer to the general comment for a response to this point.
>
> > Lines 511-515 mention that deeper networks were trained and achieved comparable or better accuracy results; however, these results are not shown. The paper only includes experiments with a single hidden layer.
>
> We have added an appendix section summarizing our preliminary results on deeper networks.
>
> > The algorithm in Function 2 (Appendix A.1) requires quantities labeled 'isx_redundancies' and 'pid_atoms' (lines 732 and 733), but it is not specified how these quantities are computed.
>
> The isx_redundancies are computed from the given probability mass function according to the analytical definition of $I_\cap^{\mathrm{sx}}$ defined in Makkeh et al. 2021, while the pid_atoms are computed from these redundancies by means of a Moebius inversion. We expanded the pseudocode section in the paper to incorporate this information.
>
> > According to Appendix A.3, the proposed method requires a lot of computational power even for MNIST. This raises concerns about its scalability to more complex datasets.
>
> We believe that optimization of the implementation as well as utilization of architectures taylored to the problems at hand (e.g., convolutional layers for image classification) will allow for more complex tasks to be solvable by infomorphic neurons. Nevertheless, we propose infomorphic networks primarily as a research tool to understand the local information-theoretic goals and suggest to use the insights gained from this analysis in the design of more computationally efficient local learning rules which implicitly optimize the same PID goals for actual applications, which is a subject of future research.

---

> ### Author Response · Authors · 2024-11-21
>
> > Is the activation function in line 283 commonly used? How the $\alpha_i$ and $\beta_i$ (for $i=1,2$) values are determined for this activation function?
>
> The activation function in line 283 is an extension of the activation function originally devised by Kay and Phillips. While we have not made a thorough investigation into the effect of the exact choice of parameters for this goal function, we believe that as long as the feedforward and lateral inputs retain their dominant effect on the outputs, the results will likely be very similar. This invariance is demonstrated by the results of the simple linear goal function suggested in Section 4.
>
> > Regarding lines 293-297, which mention that lateral connections introduce recurrence, how did you determine that presenting the same input twice is sufficient? For more complex datasets, more iterations may be needed. I am curious if the output converges after two presentations.
>
> Thank you for this valuable suggestion. Intuitively, we expect a swift convergence of the activations due to the subordinate effect of the lateral connections in the activation function. Nevertheless, in order to validate that the activation values do in fact converge after only two iterations, we conducted addional experiments which confirm that the activations indeed only marginally change after two iterations. We added a brief section to the appendix where we discuss those results as well as the reasons for the fast convergence.
>
> > Is there any particular heuristic or rationale for using 20 equally sized bins in the algorithms in Appendix A.1?
>
> The number of bins needs to be balanced between two opposing criteria: Too small bin numbers lead to a greater information loss as more different samples are gathered into the same bin. On the other hand, too large bin numbers make the information-theoretic quantities more difficult to estimate reliably. The number of 20 bins has been hand-chosen to balance between these two desiderata, although the effect of choosing slightly fewer or slightly more bins has not yet been conclusively analyzed. We have added an appendix section showcasing and discussing the effects of changing the number of bins.

---

> > ### Comment · Reviewer_wfkU · 2024-11-25
> >
> > I thank the authors for their detailed explanations and revisions. Based on their rebuttal and overall response, I have increased my score from 5 to 6.

---

### Official Review · Reviewer_dmEh · 2024-11-01

**Soundness:** 3
**Presentation:** 3
**Contribution:** 3
**Rating:** 8
**Confidence:** 4

**Summary:**

The paper presents a local learning framework for neural networks by introducing a local training objective inspired by partial information decomposition (PID) in information theory.
It trains each neuron individually by a local goal function that is a linear combination of PID atoms, which decomposes the information that three input signals
(feedforward, context, and lateral) provide about the neuron's output. The weights of the combination can be chosen heuristically or optimized on validation data. The weights represent
how information in the inputs contribute to the neuron's output and are human-interpretable. Experiments on a single-hidden-layer network show that the proposed learning method
can achieve similar performance to conventional training by cross-entropy loss, yet providing interpretable learning process of individual neurons.

**Strengths:**

The idea of using information-theoretic loss functions to train individual neurons is interesting, with nice connection to biological neural networks and neuroscience.

The novelty of adding a third input class representing lateral connections between neurons is intuitive and interesting and demonstrates better empirical performance than the bivariate model.

The overall method and motivation are clearly presented.

The experiments are well designed. Nice ablation study on the goal parameters. Good interpretation of "important" PID atoms identified by goal parameter optimization.

**Weaknesses:**

Some parts of the methodology are not clear. Please see questions below.

The PID-based goal functions and infomorphic neurons were originally proposed by Makkeh et al. 2023, and the contribution of this work is to introduce lateral connections as a third input class.
However, in the Introduction, the authors claim the PID goal function to be one of the main contributions of this paper. The authors should explain more clearly the difference between this work and Makkeh et al. 2023,
and define their contributions more accurately.

Experiments were performed on neural nets with single hidden layer, which limits the scope of the paper. It is unclear whether the observations and insights can generalize to deeper neural networks.

**Questions:**

Questions related to numerical computation of PID atoms:
- To compute the PID atoms during training, the authors empirically evaluated the joint probability mass function of the aggregated inputs and output of each neuron. Since the input is high-dimensional, is a large batch size
needed for such numerical estimation to be accurate and stable? For example, the input dimension is at least 28^2 = 784 for MNIST, and each dimension is discretized to 20 levels. Is the batch size of 1024 sufficient?

- The authors mentioned that the shared-exclusion redundancy and thus the PID atoms are differentiable with respect to the probability distribution. However, in my understanding, the empirical probability mass function
of a discrete random variable is not differentiable w.r.t. its samples. Then the goal function will not be differentiable w.r.t. a neuron's weights, which is needed for training. Could the authors clarify this point?
In particular, how is the empirical probability mass function differentiated w.r.t. the output (and subsequently the weights) of the neuron?

There seems to be a gap between the claimed "neuron-level interpretability" and the proposed learning framework.
Usually, interpretability means understanding how a trained network makes predictions and what the network has learned.
However, the method proposed in this paper only provides interpretation for the learning objective, rather than insight of what the model and each neuron has learned.
Furthermore, the interpretability is still global, not neuron-level, as the goal parameters are shared across all neurons, i.e., the interpretation for all neurons are the same.
Finally, the cross-entropy loss and the backpropagation process are quite human interpretable, in my opinion.
Could the authors comment on this?

"For the supervised classification task at hand, the function ... has been chosen ... This ensures that the network performs similarly during training, when context and lateral inputs are provided, and for evaluation,
where the context signal is withheld."
It seems that the given activation function is for training, since it uses the context (label) as input. Can the authors define clearly what the activation function during testing is?

"One promising path towards constructing deeper networks is using stacked hidden layers that receive feedback from the next layer, similar to setup 3."
It is a bit unconvincing to replace label feedback by feedback connection from the next layer. In the latter, the neuron's learning goal is to capture the part of the feedforward signal that agrees with the next layer's output, which is less intuitive than
capturing the part that agrees with the label.

Figure 8, comparision with bivariate model: Since the goal is to compare trivariate with bivariate, it might be better to put trivariate and bivariate on the same axis. If it'd be too crowded,
maybe group the results by "Heuristic"/"Optimized".

In "Goal parameters" paragraph in Experiments, it might be better to state that "heuristic goal function" is $\Pi_{ \{ F\} \{C\}}$. Although it can be inferred from Fig. 4, it's better to define it in the text as well.

In Fig. 4, is the difference in validation accuracy defined as "after setting to 0 - before"?

---

> ### Author Response · Authors · 2024-11-21
>
> > The PID-based goal functions and infomorphic neurons were originally proposed by Makkeh et al. 2023, and the contribution of this work is to introduce lateral connections as a third input class. However, in the Introduction, the authors claim the PID goal function to be one of the main contributions of this paper. The authors should explain more clearly the difference between this work and Makkeh et al. 2023, and define their contributions more accurately.
>
> The main contribution of this paper is the introduction and study of trivariate PID goal functions, which, while building on the ideas of Makkeh et al., represent a pivotal step which makes infomorphic neurons solve classification tasks better than logistic regression and achieve performance on par with backpropagation. In our research, it has emerged that three different input signals are crucial to local learning: A receptive signal providing the input, a relevance signal helping to filter this input and a distribution signal enabling self-organization between neurons.
> We agree with you that the wording in the introduction can be improved to highlight this fact and have made changes to the manuscript accordingly.
>
> > Experiments were performed on neural nets with single hidden layer, which limits the scope of the paper. It is unclear whether the observations and insights can generalize to deeper neural networks.
>
> Please refer to the general comment for a response to this point.
>
> > To compute the PID atoms during training, the authors empirically evaluated the joint probability mass function of the aggregated inputs and output of each neuron. Since the input is high-dimensional, is a large batch size needed for such numerical estimation to be accurate and stable? For example, the input dimension is at least 28^2 = 784 for MNIST, and each dimension is discretized to 20 levels. Is the batch size of 1024 sufficient?
>
> Please refer to the general comment for a response to this point.
>
> > The authors mentioned that the shared-exclusion redundancy and thus the PID atoms are differentiable with respect to the probability distribution. However, in my understanding, the empirical probability mass function of a discrete random variable is not differentiable w.r.t. its samples. Then the goal function will not be differentiable w.r.t. a neuron's weights, which is needed for training. Could the authors clarify this point? In particular, how is the empirical probability mass function differentiated w.r.t. the output (and subsequently the weights) of the neuron?
>
> In line with the approach by Kay and Phillips (1997) and Makkeh et al. (2023), the PID atoms are only differentiated with respect to the conditional probabilities of the neuron's output $Y$. The joint probability mass is constructed in two steps: First, an empirical histogram is created from the realizations of the quantized aggregated variables $F$, $L$ and $C$. For each sample, the infomorphic neuron defines the conditional probability of the neuron's binary output $Y$ given the inputs F, L and C directly, which allows to construct the full probability as $p(f, l, c, y) = p(f, l, c)p(y|f, l, c)$. While it is true that $p(f, l, c)$ is not differentiable with respect to the sources $F$, $L$ and $C$, the conditional probability $p(y|f, l, c)$ varies smoothly with the inputs, making the full probabilitiy distribution differentiable under the assumption that $p(f, l, c)$ remains constant. Despite the fact that we do not currently have a strong a priori argument why gradients of $p(f, l, c)$ can be omitted, the actual changes in PID atoms over training and the good task performance when using these gradients in experiments provides an ex post justification for this procedure. Nevertheless, in ongoing research, we are investigating the possibility of making the full pmf differentiable by means of stochastic quantization of the inputs. We clarified these points in the appendix of the manuscript.

---

> ### Author Response · Authors · 2024-11-21
>
> > There seems to be a gap between the claimed "neuron-level interpretability" and the proposed learning framework. Usually, interpretability means understanding how a trained network makes predictions and what the network has learned. However, the method proposed in this paper only provides interpretation for the learning objective, rather than insight of what the model and each neuron has learned. Furthermore, the interpretability is still global, not neuron-level, as the goal parameters are shared across all neurons, i.e., the interpretation for all neurons are the same. Finally, the cross-entropy loss and the backpropagation process are quite human interpretable, in my opinion. Could the authors comment on this?
>
> Please refer to the general comment for an explanation of how the notion of interpretability that infomorphic neurons offer is on a different abstraction level than task-level interpretability. While the goal parameters are currently shared by all neurons, all information that the neuron requires for training is provided locally and not by a global backpropagation signal, which is why this description gives a better insight into the actual information processing necessary at a local scale to fulfill a global goal in  a self-organized fashion. Investigating whether giving neurons of the same layer different goal functions can improve overall performance remains an open research question. While cross-entropy loss gives a good intuition on the mechanistic goal of the output layer's neurons, it remains opaque what information-theoretic goals the neurons of earlier layers need to fulfill in order to provide the best intermediate representation to the output layer.
>
> > "For the supervised classification task at hand, the function ... has been chosen ... This ensures that the network performs similarly during training, when context and lateral inputs are provided, and for evaluation, where the context signal is withheld." It seems that the given activation function is for training, since it uses the context (label) as input. Can the authors define clearly what the activation function during testing is?
>
> During testing, the same activation function is used, but the context input is set to zero. We have added a clarification for this in the manuscript.
>
> > "One promising path towards constructing deeper networks is using stacked hidden layers that receive feedback from the next layer, similar to setup 3." It is a bit unconvincing to replace label feedback by feedback connection from the next layer. In the latter, the neuron's learning goal is to capture the part of the feedforward signal that agrees with the next layer's output, which is less intuitive than capturing the part that agrees with the label.
>
> The idea to use feedback connections as context instead of the full label is to achieve more biological plausibility and locality by not providing the full label information to every hidden layer. The idea is for the network to train hierarchical representations "from back to front", the later layers informing the earlier layers which important information to forward.
> Nevertheless, we agree with the reviewer that this idea needs further development and providing the full label to all hidden layers marks the most straightforward and promising generalization to multiple hidden layers for now. We have included an appendix section which summarizes our preliminary results on this topic, showing that deeper infomorphic networks successfully train with the same goal functions as their shallow counterparts and achieve similar performance for fewer trained weight parameters.
>
> > Figure 8, comparision with bivariate model: Since the goal is to compare trivariate with bivariate, it might be better to put trivariate and bivariate on the same axis. If it'd be too crowded, maybe group the results by "Heuristic"/"Optimized".
>
> We agree that this change facilitates the most important comparisons and have made the suggested changes to Figure 8.
>
> > In "Goal parameters" paragraph in Experiments, it might be better to state that "heuristic goal function" is $\Pi_{\{F\}\{C\}}$. Although it can be inferred from Fig. 4, it's better to define it in the text as well.
>
> The heuristic trivariate goal function for classification tasks is introduced in the last paragraph of Section 3 as the combination of two bivariate goal functions. We have highlighted this section better in the manuscript.
>
> > In Fig. 4, is the difference in validation accuracy defined as "after setting to 0 - before"?
>
> Yes, both Figures 4B and 4D show the same concept of difference in validation accuracy for setting goal function parameters to zero for different learning tasks.

---

> ### Comment · Reviewer_dmEh · 2024-11-25
>
> Thank the authors for their responses.
>
> Can the authors elaborate how p(y | f, l, c) is computed? The authors state that the loss can be differentiated w.r.t. p(y | f, l, c) - assuming p(f, l, c) is constant - and p(y | f, l, c) can be differentiated w.r.t. the neuron weights. However, p(y | f, l, c) is not dependent on the weights, because Y is directly given by the activation function defined as A(F,C,L) = F[(1 − α1 − α2) + α1 σ(β1FC) + α2 σ(β2FL)], where alpha and beta are all fixed. Can the authors clarify if I have any misunderstanding? Specifically, how is p(y | f, l, c) and its gradient w.r.t. the weights computed?

---

> > ### Author Response · Authors · 2024-11-27
> >
> > You are correct in assessing that the conditional probability $p(y\mid f,c,l)$ is defined directly from the activation function. Specifically, the probabilities for the two possible outcomes, $y=1$ and $y=−1$, are given by $p(y=1\mid f,c,l)=\sigma(A(f,c,l))$ and $p(y=−1\mid f,c,l)=1−\sigma(A(f,c,l))$, respectively. Note, however, that these probabilities depend on the weights, as $f$, $c$, and $l$ represent weighted sums of the high-dimensional inputs $\mathbf{X}_F$, $\mathbf{X}_C$ and $\mathbf{X}_L$ of the three sources, respectively. Consequently, the gradients with respect to these weights can be computed straightforwardly using the chain rule.

---

> > > ### Comment · Reviewer_dmEh · 2024-11-27
> > >
> > > Thank you for your response.
> > >
> > > I presumed that you computed $p(y \mid f, c, l)$ for each bin, but it seems that you computed it for each sample. I believe $p(y \mid f, c, l)$ for each bin is needed to calculate the atoms in information decomposition. Could you clarify how $p(y \mid f, c, l)$ for each sample is converted to $p(y \mid f, c, l)$ for each bin?

---

> > > > ### Author Response · Authors · 2024-11-28
> > > >
> > > > Thank you for your interest in the implementation details, which helps us make our methodology clearer.
> > > >
> > > > To compute the PID, the joint probability masses $p(y, \hat{f}, \hat{c}, \hat{l})$ for each bin, i.e., for all combinations of $y$ and the binned $\hat{f}$, $\hat{c}$ and $\hat{l}$, are required (i.e., 2x20x20x20 values in total, of which, however, many are zero). To convert the conditional probability $p(y \mid f, c, l)$ for each sample to $p(y \mid \hat{f}, \hat{c}, \hat{l})$ for each bin, the per-sample values $p(y \mid f, c, l)$ where $(f, c, l)$ fall into the same bin $(\hat{f}, \hat{c}, \hat{l})$ are averaged. In our implementation, $p(y, \hat{f}, \hat{c}, \hat{l})$ is constructed as a weighted histogram of size 2x20x20x20. For each sample $(f, c, l)$, the probability $p(y\mid f, c, l)=\sigma(A(r,c,l))$ is computed, after which each sample is added to the histogram twice: Once into the bin $(y=+1, \hat{f}, \hat{c}, \hat{l})$ with a weight of $p(y=+1\mid f, c, l)$ and once to the bin $(y=-1, \hat{f}, \hat{c}, \hat{l})$ with weight $p(y=-1 \mid f, c, l)=1-p(y=+1\mid f, c,l)$. Finally, the histogram is normalized by dividing it by the number of samples in the batch, producing $p(y, \hat{f}, \hat{c}, \hat{l})=p(y\mid f, c, l)p(\hat{f}, \hat{c}, \hat{l})$, where $p(\hat{f}, \hat{c}, \hat{l})$ is obtained implicitly from the frequency of samples falling into the corresponding bin.

---

> > > > > ### Comment · Reviewer_dmEh · 2024-11-30
> > > > >
> > > > > Thank the authors for their detailed explanation and addressing all my questions. I have increased my score from 6 to 8.

---

### Official Review · Reviewer_Mc5F · 2024-11-02

**Soundness:** 4
**Presentation:** 4
**Contribution:** 3
**Rating:** 8
**Confidence:** 5

**Summary:**

Based on the concept of partial information decomposition in the information theory, this work formulates a learning framework that allows the implementation of per-neuron goal functions and neuronwise interpretability. In an ANN setup for classification tasks, the goal function is optimized via parameters of the ‘informorphic’ neurons, with neuron-level interpretation, good performance (comparable to the same ANN trained with bp) and insights into the local computational goals. Examples with bivariate and trivariate informorphic neurons are demonstrated where the three-input classes unlock the potential of information theoretic learning, validating the abovementioned advantages in comparison with classical information composition.

**Strengths:**

The use of partial information decomposition in a learning framework with bivariate and trivariate implementations enables interpretable information processing at the per-neuron level, which is also new. The discussion based on a comparison between heuristic and optimization approaches demonstrates the potential of the interpretability of local learning framework in a task-relevant context and without the loss of performance compared to ANN trained with bp.

**Weaknesses:**

The title ‘What should a neuron aim for’ is broader than the scope explored in the current work, where a single-layer bivariate and trivariate local learning framework is studied. There is still a large gap between the interpretability of the model here and that of the neuron. The advantage of interpretability from an application perspective is not demonstrated or discussed, which is expected to be very limited by the single-layer simplicity here.

**Questions:**

How do we understand the performance presented in Fig. 3B by considering that at Nhid = 100, there is a convergence for all results? Why a sparse lateral outperform a dense lateral under large Nhid conditions? What are the computational and memory costs?
What does the extraction of high-effect parameters mean, and would this be useful to construct networks with higher performance, or understand the nature of the problems under training?

---

> ### Author Response · Authors · 2024-11-21
>
> > The title ‘What should a neuron aim for’ is broader than the scope explored in the current work, where a single-layer bivariate and trivariate local learning framework is studied. There is still a large gap between the interpretability of the model here and that of the neuron. The advantage of interpretability from an application perspective is not demonstrated or discussed, which is expected to be very limited by the single-layer simplicity here.
>
> Please note that, as outlined in the general comment, the more abstract, information-theoretic notion of interpretability that infomorphic neurons provide differs from other notions of interpretability of networks solving a specific task.
> While we believe that the trivariate infomorphic framework has the potential to uncover 'what a neuron should aim for' also in more complex networks, we agree that the title may not optimally reflect the scope of this paper in particular. The most important contribution of this paper is the introduction of trivariate PID goal functions, which incorporate an additional lateral input that allows for neurons to self-organize to encode unique relevant information contributions. Highlighting this self-organization aspect, we thus propose "Neuron, do your part: Self-organizing global computation from local objective functions based on partial information decomposition" as the new title of this work. Please let us know if you find this title adequate or whether you would prefer shortening the original title to "Designing neuron-local objective functions based on information theory", staying closer to the original submission.
>
> > How do we understand the performance presented in Fig. 3B by considering that at Nhid = 100, there is a convergence for all results?
>
> As outlined in the legend of Fig. 3B, the hyperparameters of the goal function have been optimized only once for $N_\mathrm{hid}=100$ neurons and then reused for different values of $N_\mathrm{hid}$. Thus, it is expected that the infomorphic networks with $N_\mathrm{hid}=100$ neurons come closest to the backpropagation performance, since the hyperparameters may not be strictly optimal for other values of $N_\mathrm{hid}$. Furthermore, the networks with sparse connectivity have the number of connected neurons limited to at most 100, meaning that it is equivalent to the fully connected case for $N_\mathrm{hid} \leq 100$. We highlighted this fact better in the caption of Figure 3.
>
> > Why a sparse lateral outperform a dense lateral under large Nhid conditions?
>
> Why exactly the sparse lateral connections outperform their dense counterparts and what the optimal sparsity level is remains an ongoing area of research. Our best current hypothesis is that once the network layer becomes large enough to encode the relevant label information many times over, it may no longer be optimal for each neuron to strive for unique information with respect to all other neurons. Instead, weakening the uniqueness constraint by considering only a subset of lateral neurons may---especially given the stochastic neuron outputs---lead to more robust representations.
>
> > What are the computational and memory costs?
>
> Please refer to Appendix A.3 for an approximate estimation of the compute ressources used in this project.
>
> > What does the extraction of high-effect parameters mean, and would this be useful to construct networks with higher performance, or understand the nature of the problems under training?
>
> The parameter importance analysis reveals that the size of the PID goal function parameters alone does not necessarily reflect their significance in the training process, which gives additional insights into which kind of information processing is important to optimize for at the neuron level. In the future, these insights may indeed be used to prune low-effect goal function components to achieve higher efficiency. Furthermore, these insights may aid comparisons to the PID footprint of classical non-infomorphic local learning rules.

---

> > ### Comment · Reviewer_Mc5F · 2024-11-30
> > **Comments on the authors replies**
> >
> > I would like to thank the authors for clarifying the points I mentioned in the original reviewer report, which addressed all of my concerns.

---

### Official Review · Reviewer_SbrR · 2024-11-03

**Soundness:** 3
**Presentation:** 3
**Contribution:** 4
**Rating:** 8
**Confidence:** 3

**Summary:**

The authors claim that modern deep neural networks focus on global optimization and the learning objective for individual neurons are obscure, whereas real-life biological neurons grow with local learning objectives with limited global information. Based on this intuition, the authors propose a biologically inspired neural network that enhances interpretability of the learning dynamics of individual neurons. The authors leverage Partial Information Decomposition (PID) in information theory to formulate objective functions on the individual neuron level, and these neurons are termed “infomorphic neurons”. This novel formulation is able to achieve comparable performance to backpropagation on MNIST and CIFAR10 datasets, showing preliminary signs of utility. Besides, the proposed framework allows for interpretability analysis on the partial information components which is quite unique and interesting.

**Strengths:**

1.	The topic of biologically inspired neural networks has attracted great attention in the recent years. Proposing alternative solutions to the long-dominating backpropagation method is also interesting and profoundly influential. I encourage the authors to work along these lines.
2.	Overall this is a paper very rich in content.
3.	The visualizations in Figure 1 and 2 are very helpful for understanding the partial information components in partial information decomposition.
4.	The performance on MNIST and CIFAR10 are quite promising. From what I read, the proposed method achieved beyond 98% accuracy on MNIST and 94.4% accuracy on CIFAR10.
5.	The parameter importance assessment is thoughtful and informative.

**Weaknesses:**

1.	While I believe we should not expect super comprehensive experiments from a paper that initially introduce a novel concept or paradigm, it might be even improve the soundness of this paper if the authors can also include the performance on slightly more challenging datasets. If standard large datasets such as ImageNet are too computationally expensive, the authors could consider variants that are decently big and challenging, such as TinyImageNet.
2.	For demonstration of experimental results, while I like the richness of Figure 3B and I do notice Figure 8 in the appendix, I would personally argue it would be more straightforward to include a “less exiting” bar and whiskers plot of backprop as well as different variants of bivariate and trivariate models. I suspect the authors left the performance of the bivariate model to the appendix to avoid overcrowding the results, but this omission from the main text might trigger confusion from the audience. Again, a good old bar and whisker might be a valid solution.

**Questions:**

1.	Please refer to Weakness 2. Would the authors consider the “less existing” representation of the quantitative results or provide some other alternative that are similarly straightforward?
2.	I still have a hard time understanding how the partial information components are realized in actual implementation. It would be good for the authors to provide a brief explanation on how that is done or point to the relevant text in the paper.
3.	It is a bit uncommon to have “Related Works” and “Limitations and Outlook” as part of the Discussion section. Is that intentional or is it just a typo?

---

> ### Author Response · Authors · 2024-11-21
>
> We first want to raise your attention to an inaccuracy in your review: As stated in the paper, infomorphic networks achieve a test set accuracy of 97.5% on MNIST (for 2000 neurons and sparse connections) and 42.5% on CIFAR10 (for only 100 neurons), the latter matching or slightly outperforming backpropagation training on the same stochastic binary-activation networks.
>
> > While I believe we should not expect super comprehensive experiments from a paper that initially introduce a novel concept or paradigm, it might be even improve the soundness of this paper if the authors can also include the performance on slightly more challenging datasets. If standard large datasets such as ImageNet are too computationally expensive, the authors could consider variants that are decently big and challenging, such as TinyImageNet.
>
> We agree that validating our findings on more complex tasks is an important next step to establish infomorphic networks. However, our results indicate that in order to solve more complex tasks, more hidden layers and taylored architectures (e.g. convolutional layers) are required, which are subjects of ongoing research. As mentioned in the gerenal comment, however, we have since applied infomorphic networks successfully to the AudioMNIST task, demonstrating the applicability of our approach for tasks beyond image classification.
>
> > For demonstration of experimental results, while I like the richness of Figure 3B and I do notice Figure 8 in the appendix, I would personally argue it would be more straightforward to include a “less exiting” bar and whiskers plot of backprop as well as different variants of bivariate and trivariate models. I suspect the authors left the performance of the bivariate model to the appendix to avoid overcrowding the results, but this omission from the main text might trigger confusion from the audience. Again, a good old bar and whisker might be a valid solution.
>
> We agree that a simple overview of the network performances of all different setups for a fixed number of $N_\mathrm{hid} = 100$ neurons facilitates the comparison between the different setups. We have thus added a new subplot to Figure 3 which shows all bivariate, trivariate and backprop results.
>
> > I still have a hard time understanding how the partial information components are realized in actual implementation. It would be good for the authors to provide a brief explanation on how that is done or point to the relevant text in the paper.
>
> The computation of the PID atoms works as follows: First, the aggregated input variables $F$, $L$ and $C$ are computed as weighted sums of the feed-forward, lateral and context inputs. Subsequently, these scalars are quantized to 20 levels each, and together with the known conditional probabilities of the stochastic binary output given the inputs the joint probability mass function is constructed from a batch of 1024 samples. From this probability mass function, the generalized redundancies $I^\mathrm{sx}_\cap$ are computed according to the analytical definition by Makkeh et al. 2021. Since these redundancies can be written as sums of PID atoms, the PID atoms themselves can finally be calculated from the redundancies by a Moebius inversion (i.e., generalized inclusion-exclusion-rule) of this lattice structure. In our manuscript, this procedure is explained in Chapter 4 together with the pseudocode provided in Appendix A.1. We have extended the pseudocode to explain how the redundancies and PID atoms are computed to make this point more clear.
>
> > It is a bit uncommon to have “Related Works” and “Limitations and Outlook” as part of the Discussion section. Is that intentional or is it just a typo?
>
> Thank you for this notice. We have reorganized the last sections to adhere to a more standard structure.

---

> > ### Comment · Reviewer_SbrR · 2024-11-24
> > **Response to rebuttal**
> >
> > 1. Thanks for pointing out the incorrect quote in my original review. I misread the numbers.
> > 2. Thanks for updating Figure 3 and including the more straightforward comparison.
> > 3. Thanks for explaining the PID implementation.
> >
> > Best of luck!

---

### Author Response · Authors · 2024-11-21
**General Response to the Reviewers**

Thank you for your insightful and constructive reviews. Incorporating your suggestions into our manuscript has certainly helped to improve the clarity and precision of our work.
To avoid duplication, we will address questions raised by multiple reviewers in this general comment.

**Dimensionality of inputs and estimation of PID atoms**

Two reviewers raised the question if the dimensionality of the input or the batch size of 1024 pose a problem to the estimation of the PID atoms. We believe this not to be the case, as the high-dimensional input or lateral connections are not used in the estimation themselves but only the aggregated scalar values R, C and L which are reduced to a single dimension each by a learned weighted sum. For this reason, the joint probability mass function is only four-dimensional irrespective of the dimensions of the inputs. Furthermore, the probability mass function has empirically been observed to be quite sparse, making 1024 samples sufficient to give a sufficiently good approximation of the PID atoms.
We validate this assumption, we ran MNIST classifier networks with $N=100$ neurons in the hidden layer with smaller and larger batch sizes. Despite matching the learning rates accordingly, in this first test, the runs with larger batch sizes converge significantly slower but reach approximately the same final accuracy (see new appendix section). While determining the optimal batch size remains a question for future research, we thus believe the chosen batch size of 1024 samples to be sufficient for the experiments shown.

**Scaling to multiple hidden layers for solving more complex tasks**

Furthermore, multiple reviewers have highlighted the importance of analyzing multi-layer networks using infomorphic neurons. We agree that using deeper layers is the logical next step in our research and have expanded on the preliminary results in a new appendix section. These preliminary results show that deeper infomorphic networks can train with the same goal functions as their more shallow counterparts, reaching comparable or slightly higher accuracy when matching the number of learned parameters. Nevertheless, since the goal of this paper is to introduce the framework of trivariate infomorphic networks, a thorough investigation of how optimal goal functions differ between hidden layers is left to future research.
To showcase the generality of PID goal functions, we have in the meantime applied our approach to the AudioMNIST dataset, achieving test accuracies between $94.7\%$ and $96.3\%$ for 10 training runs using the same optimized goal function from the MNIST image recognition task. While the AudioMNIST task may not be significantly more complex, this result showcases the transferability of the local goal function to classification tasks from domains other than image recognition, which likely have very different input statistics.

**On the concept of "interpretability"**

We want to emphasize that the more abstract notion of task-independent interpretability of local goals provided by infomorphic networks is  different from interpretability from an application perspective: The interpretability that infomorphic neurons offer is on the level of the information-theoretic goals of the individual neurons, revealing what information processing on the local level is sufficient to solve a particular global task. These results are expected to depend on the type of task (e.g. classification), but should be transferrable between tasks of the same nature (e.g. MNIST and CIFAR10) and between different mechanism which produce the activation values (e.g., activation function).
In future work, these tools may be used to discover which neurons do or do not contribute to the solving of a global task. Because the used redundancy measure is *local*, meaning it can be evaluated for individual samples, infomorphic neurons may in future work be used to identify particular classification labels for which a neuron does or does not contribute.
We added a paragraph to the paper's discussion to explain this distinction.

---

### Author Response · Authors · 2024-12-02
**Manuscript Title Change**

Following a suggestion by reviewer Mc5F, we have previously changed the title of our work to "Neuron, do your part: Self-organizing global computation from local objective functions based on partial information decomposition" to address the concern that the question "What should a neuron aim for?" in the original title  may be understood to imply more extensive analysis of larger networks. However, in communication with colleagues we found that the new title can be misunderstood as implying a global agent that "makes neurons do their part", when focus should be on their local learning and self-organization.

Should the reviewers and editors agree, we therefore suggest to revert to the original title "What should a neuron aim for? Designing local objective functions based on information theory". We believe that the first part of this title introduces the topic in a thought-inspiring manner and highlights the generality of the proposed framework in principle, with the second half clearly defining the scope of the present submission. If the reviewers have serious concerns with this original paper title, or if regulations disallow further changes of the title at this point, we are are nevertheless contented with keeping the new suggested title.

The regulations regarding title changes remain unclear to us: While the ICLR author guidelines clearly specify that the title can be augmented during the rebuttal phase, there was no option to update the official title in the rebuttal form on the OpenReview website. We thus kindly ask the editor to clarify which title should be used in the camera-ready version in accordance with the regulations in their final decision.

---

### Meta-Review · Area_Chair_4eVi · 2024-12-07

**Metareview:**

This work deals with the dissonance between global and local learning rules in neural networks. It takes an information theory approach to enable neurons to steer the integration of different classes of input information to drive it's own update. The way that the different classes of information can be combined can be dictated by the researcher or fit to data. The reviewers noted that the overall conceptual framework was novel and interesting, and especially appreciated the interpretable nature of the model parameters.

While some minor concerns were raised, these mostly focused on details, e.g., the application to harder problems and questions on the data-intensive nature of information estimation. A successful discussion with reviewers and additional preliminary results on new experiments seem to have overcome these concerns.

**Additional Comments On Reviewer Discussion:**

In response to the reviewer comments, new experiments were run. I do not see concern for those experiments as the core of the model was not changed in response to reviewers, just additional validation.

---

### Decision · Program_Chairs · 2025-01-22

Accept (Oral)